



# 1 Spatial and temporal distribution of fine aerosol acidity in the
# 2 Eastern Mediterranean

Anna Maria Neroladaki[1,2], Maria Tsagkaraki[1], Kyriaki Papoutsidaki[1], Kalliopi Tavernaraki[1], Filothei
Boufidou[3], Pavlos Zarmpas[1], Irini Tsiodra[4], Eleni Liakakou[4], Aikaterini Bougiatioti[4], Giorgos
Kouvarakis[1], Nikos Kalivitis[1], Christos Kaltsonoudis[2], Athanasios Karagioras[5], Dimitris Balis[6],
Konstantinos Mihailidis[6], Konstantinos Kourtidis[5], Stelios Myriokefalitakis[4], Nikos Hatzianastassiou[7],
Spyros N. Pandis[2], Athanasios Nenes[2,8], Nikolaos Mihalopoulos[1,3] and Maria Kanakidou[1,2,7]
[1]Environmental Chemical Processes Laboratory (ECPL), Department of Chemistry, University of Crete, 70013
Heraklion, Greece
[2]Center for Studies of Air Quality and Climate Change, Institute for Chemical Engineering Sciences, Foundation
for Research and Technology Hellas, 26504 Patras, Greece
[3]Institute of Environmental Physics, University of Bremen, 28359 Bremen, Germany
[4]Institute for Environmental Research and Sustainable Development, National Observatory of Athens, 15236
Palea Penteli, Greece
[5] Department of Environmental Engineering, Democritus University of Thrace, 67100 Xanthi, Greece
[6]Laboratory of Atmospheric Physics, Aristotle University of Thessaloniki, 54634 Thessaloniki, Greece
[7]Laboratory of Meteorology, Department of Physics, University of Ioannina, 45110 Ioannina, Greece
[8]Laboratory of Atmospheric Processes and their Impacts, School of Architecture, Civil and Environmental
Engineering, Ecole Polytechnique Fédérale de Lausanne, 1015, Lausanne, Switzerland
*Correspondence to*: Maria Kanakidou (mariak@uoc.gr)
**Abstract.** Aerosol acidity (pH) affects aerosol composition and properties, and therefore climate, human health
and ecosystems. Fine aerosol acidity and its seasonal variation at 6 sites (Finokalia, Patras, Thissio, Ioannina,
Thessaloniki, and Xanthi) in Greece were investigated during 2019-2020. The thermodynamic model
ISORROPIA-lite was used to calculate aerosol water and acidity based on measurements of the chemical
composition of $PM_{2.5}$ and available gas-phase concentrations of $HNO_3$, $NH_3$, and HCl. During winter the fine
aerosols were acidic to moderately acidic throughout Greece with an overall mean aerosol pH of 3.57±0.44 in
urban areas and 3.05±0.50 in remote locations. The highest aerosol pH (4.08±0.42) in January 2020 was found in
Ioannina due to, among others, high $K^+$ levels from biomass burning emissions. Aerosols in Xanthi were the most
acidic due to high sulfate levels. Similar seasonal profiles of aerosol pH were observed at all sites studied with
different factors contributing to this seasonality. During the summer $PM_{2.5}$ at Thissio, Ioannina and Finokalia was
acidic with a mean aerosol pH across all three sites of 1.76±0.40. During this season, sulfates were the driver of
the higher acidity conditions at Thissio and Finokalia, with other factors such as the semivolatiles and temperature
contributing to a lesser extent. At Ioannina, temperature along with the total ammonia and nitrate were the main
contributors to the seasonal difference of the aerosol pH, while some of the nonvolatile species also contributed.
In most cases, the importance of organics for aerosol pH was small.
**1 Introduction**



The aerosol pH is one of the most important chemical properties of aerosols. It controls the rates of several reactions in the particulate phase (e.g., sulfate and oligomer formation reactions in the particulate phase) and governs the gas-particle partitioning of semivolatile gases such as ammonia ($NH_3$), nitric acid ($HNO_3$) and hydrochloric acid (HCl), and some organic acids with low molecular weight (formic, oxalic, acetic etc.) and bases (e.g. amines). Several studies have thoroughly discussed the importance of the gas phase $NH_3$ and $HNO_3$ and their effect on the particulate matter as well as their relationship with aerosol pH and water (Pye et al., 2020; Nenes et al., 2020; Guo et al., 2018, 2016, 2017b). $NH_3$ from anthropogenic and biogenic emissions is the most important gaseous base in the atmosphere, while $HNO_3$ is an acid produced by the oxidation of $NO_x$ mainly emitted by combustion sources (Seinfeld and Pandis, 2006). $HNO_3$, because of its strong acidity and solubility, partitions more in the gas phase than in the particle phase (as nitrate) at lower aerosol pH (usually below 1.5 to 2) (Weber et al., 2016), while $NH_3$ remains in the gas phase at higher pH. These processes have serious implications for the wet and dry deposition of these semivolatile pollutants on ecosystems, their lifetime, climate and human health (Pye et al., 2020).

Metal cations, including those that originate from dust and sea salt, are nonvolatile at atmospheric temperatures. These constituents seriously affect the aerosol pH together with parameters such as the meteorological conditions (i.e. temperature and relative humidity). These parameters and the availability of sulfuric acid (that partitions almost exclusively to the particulate phase) are mostly responsible for the wide range of the pH of atmospheric particles (Pye et al., 2020).

Aerosol pH affects the uptake of sulfur dioxide ($SO_2$) into the aerosol aqueous phase and its oxidation to sulfate (Seinfeld and Pandis, 2006). This is important for climate, as sulfate is one of the main scattering components of atmospheric aerosols with a cooling effect on climate. Acid catalyzed reactions lead to the formation of brown carbon that is an important contributor to absorbing aerosol (Zhang et al., 2020). Atmospheric acidity affects aerosol toxicity, controlling the solubilization of toxic forms of transition and heavy metals, such as copper and iron (Fang et al., 2017) with adverse impacts on human health (Vierke et al., 2013). Furthermore, acidity increases the solubility and thus the bioavailability of iron, phosphorus and trace metals that are deposited on ecosystems (Pye et al., 2020). This is of a great importance in oligotrophic areas, such as the East Mediterranean Sea, a marine desert, characterized by extremely low biological productivity, due to the limited availability of nutrients like phosphate (Powley et al., 2017). The area is frequently influenced by dust outbreaks from the Saharan desert, which contain several minerals and metals primarily in insoluble forms. The conversion of insoluble to soluble bioavailable form of a given nutrient is favored under acidic conditions (Kanakidou et al., 2018; Nenes et al., 2011). The solubility of iron and thus its bioavailability to ecosystems, is also controlled by acidity, as it is enhanced in the presence of acidic species. Theodosi et al. (2008) found that in the Eastern Mediterranean the iron solubility ranged from 28% for polluted rainwater (pH 4-5) to 0.5% for Sahara dust episodes (pH=8).

Although the acidity of atmospheric particles plays such a critical role in all of the atmospheric processes mentioned above, there is no direct method for its accurate measurement due to the miniscule sample mass and liquid volume (Pye et al., 2020). Thus, several indirect methods have been developed, which estimate aerosol acidity (Pye et al., 2020). One of the best available methods is the aerosol pH prediction using atmospheric aerosol thermodynamic models in combination with measurements of aerosol composition and gas-phase nitric acid, ammonia and hydrochloric acid concentrations. Models such as ISORROPIA II (Fountoukis and Nenes, 2007),



MOSAIC (Zaveri et al., 2008), E-AIM; (Clegg et al., 2001; Wexler and Clegg, 2002), etc. can be used to this
purpose.
Aerosols' acidity covers a wide range and aerosol pH can be as high as about 7 pH units (sea salt), and as low
as -1 and sometimes even -2 pH units, depending on the composition of the aerosol and the meteorological
conditions in a given area (Pye et al., 2020). Negative values appear when sulfates are the dominant constituent
of particulate matter, while the aerosol pH rarely rises above neutral levels. Globally fine atmospheric particles
($PM_{2.5}$) have been found to have a bimodal acidity pattern with one population of smaller particles having an
average pH between 1 and 3 pH units, and another of larger particles with a mean pH close to 4 and 5 units, which
is due to the influence of dust, sea spray and biomass burning (Sindosi et al., 2012; Pye et al., 2020).
Several studies have investigated the levels of the acidity of fine particles and their major drivers at specific
sites. For instance, studies in Beijing and close-by locations indicate a mean wintertime $PM_{2.5}$ pH of 4.2 to 4.9
units (Liu et al., 2017; Guo et al., 2017a; Shi et al., 2017; Ding et al., 2019). Pye et al. (2020) summarized and
investigated the fine aerosol pH in several regions globally and reported that overall aerosol pH is consistently
acidic and looking into the seasonality, during wintertime where low temperature and high relative humidity
occur, aerosol pH was higher compared to summertime following the temperature and the availability of liquid
water content. Tao and Murphy (2019) reported that in winter the pH of fine aerosols was around 3 units at 6 sites
in Canada and was strongly influenced by temperature, relative humidity and aerosol chemical composition.
Kakavas et al. (2021) simulated the aerosol acidity and its variation with size and altitude over Europe during
summer and found the ground level mean aerosol pH across the domain to be 2.05 for $PM_1$, 2.65 for $PM_{1-2.5}$, 3.2
for $PM_{2.5-5}$ and 3.35 for $PM_{5-10}$. The lowest aerosol pH was reported in the Mediterranean and especially in the
eastern Mediterranean where high sulfate and nitrate levels were predicted. This study suggests that Eastern
Mediterranean can be an especially interesting region for aerosol pH studies.
There have been only limited estimations of aerosol acidity in the eastern Mediterranean. The $PM_{2.5}$ pH in 6
cities in the eastern Po Valley in Italy (Masiol et al., 2020) was found to range from 1.5 to 4.5 with summer
minima and winter maxima and mean pH values across all sites of 2.2 ± 0.3 and 3.9 ± 0.3, respectively. These
levels of fine aerosol acidity were mainly driven by secondary sulfate, fossil fuel combustion, secondary nitrate
and biomass burning. An earlier study in the same area (Squizzato et al., 2013) investigated seasonally the aerosol
pH and found that in spring the aerosol pH was higher compared to summer (average aerosol pH across the sites;
spring 3.6 pH units, summer 2.3, autumn 3, winter 3.6) as a result of desert dust aerosol originating from the
Sahara desert. Again in Po Valley, a multiyear (25 years) trend of fog water pH and aerosol pH was estimated
(Paglione et al., 2021) and an opposite trend between them was found; there was an increase in the fog water pH
and a decrease in fine aerosol acidity. In fact, there was a decrease of 0.5-1.5 pH units in the aerosol pH, trend
that was driven by the contemporary decrease of the corresponding air pollutants due to environmental policies in
combination with the changing meteorology (temperature and relative humidity levels). Lastly, the submicron
aerosol pH was estimated in a study conducted at the Finokalia atmospheric observation station in Greece
(Bougiatioti et al., 2016) and was found to be highly acidic ranging from 0.5 to 2.8 pH units with daytime
minimum and nighttime maximum values due to low aerosol water content and high temperatures during the day.
They also pointed out the influence of biomass burning which increased the aerosol pH values highlighting the
impact of nonvolatile cations, mainly potassium from biomass burning together with ammonia and nitrate emitted





from wood burning. Despite these studies, the spatial distribution of aerosol acidity based on atmospheric
composition observations is not well understood.
The present study aims to provide a spatial and seasonal picture of the acidity of fine (PM$_{2.5}$) atmospheric
aerosol over Greece in the eastern Mediterranean based on observations of atmospheric composition. For this, the
ISORROPIA-lite thermodynamic equilibrium model (Kakavas et al., 2022) is used together with observations of
the chemical composition of the fine aerosol from 6 sites over Greece and with gas phase NH$_3$ and HNO$_3$ data,
where available. The water and pH of the fine aerosol are estimated during summer 2019 and winter 2019-2020.
The factors controlling the seasonality of aerosol pH are examined and the effect of particulate organic matter on
aerosol pH predictions is also investigated.
**2 Measurements and Methodology**
2.1 Measurement sites
Measurements of the chemical composition of PM$_{2.5}$ were performed at 6 sites across Greece (Fig. 1) (Finokalia,
Thissio, Patra, Ioannina, Xanthi and Thessaloniki, Table S1). The corresponding summer and winter field
campaigns were conducted within the PANACEA (PANhellenic infrastructure for Atmospheric Composition and
climatE chAnge) project during the summer of 2019 and the winter of 2019-2020.
The Finokalia atmospheric observatory of the University of Crete in Crete, Greece (FKL, 35.33º N, 25.67º E;
250 m a.s.l) is a remote regional background site in the northeast coast of the island of Crete (south Greece). The
site is not subject to any major anthropogenic influence and it is considered as a representative background site
for the entire eastern Mediterranean. During the warm months of the year (April to September, dry season) the
station mainly receives air masses from the N/NW (originating from the Central and Eastern Europe and the
Balkans), while between October and April (wet season) air masses coming from the South lead to Saharan dust
events (Mihalopoulos et al., 1997).
The Thissio Air Monitoring station (THI, 37.97º N, 23.72º E, 105 m a.s.l.) of the National Observatory of
Athens, Greece is located approximately 50 m above the mean city level near the historical city center. The station
is an urban background site due to its distance from traffic and industrial emission sources and receives air
pollutants from various urban and regional sources.
The Patras site is located at the Institute of Chemical Engineering Sciences (ICE-HT) of FORTH (Foundation
for Research and Technology, Hellas), which is in Platani (PTR, 38.30º N, 21.81º E, 100 m a.s.l.) 8 km from the
city center. It is an urban background site. Local sources include transportation, biomass burning (both residential
and agricultural) and shipping emissions, while long-range transport is the dominant PM$_{2.5}$ source during most
periods (Pikridas et al., 2013).
The Xanthi station is operated by the Laboratory of Atmospheric Pollution and Pollution Control Engineering
of Atmospheric Pollutants of the Department of Environmental Engineering. The station is located in the
Kimmeria DUTH campus (XAN, 41.15º N, 24.92º E; 75 m a.s.l.) almost 2 km from the city of Xanthi in
northeastern Greece. The station is a rural site and the edge of a slope facing to the south, 20 km away from the
seashore. The Rodopi Mountain Range is located to the north of the station. The prevailing winds that reach the
site in winter are fairly stable SW/S/SE during the day, but they change to N/NW at night; the location of the site





between mountains and a valley creates a closed circulation cell of valley/land breezes (Kastelis and Kourtidis,
158     2016).

The Thessaloniki station at the north of Greece is at the Laboratory of Atmospheric Physics (LAP) which is
located at the School of Sciences of the Aristotle University of Thessaloniki (LAP, 40.63ºN, 22.95ºE; 50m a.s.l.).
Thessaloniki is a coastal city at Thermaikos Gulf and is the second largest city in Greece. The site experiences air
pollution episodes due to the meteorological conditions over northern Greece; with high pressure, anticyclonic
systems and sunny weather in summer, while colder temperatures near the surface and snow occur in winter
(Flocas et al., 2009). It is considered as a representative urban station.
Ioannina (IOA, 39.653195°N, 20.854208°E) is located in Epirus Region in northwestern Greece, which is
separated from the eastern part of the country mainland by the Pindus mountain range (orientated from NW to SE
and exceeding 2000 m in height). The sampling station was located at a kindergarten yard, 1.5 km from Ioannina's
city center. Ioannina is located on a plateau of about 500 m altitude and is surrounded by mountains. The city is
next to the Pamvotis lake and is characterized by frequent fog events in winter, due to increased relative humidity,
weak winds and basin-like attributes and winter biomass burning events (Kaskaoutis et al., 2020).
2.2 Measurements
The PANACEA campaigns took place at the 6 sites discussed above during the summer of 2019 and the winter
of 2019-2020. Atmospheric particles were collected daily on quartz-fiber filters using high-volume or low-volume
aerosol samplers depending on the site. A Sunset Organic Carbon (OC) /Elemental Carbon (EC) analyzer was
used to determine the concentrations of OC and EC, while the inorganic cations: $NH_4^+$, $K^+$, $Ca^{2+}$, $Mg^{2+}$, $Na^+$ and
anions: $SO_4^{2-}$, $Cl^-$ and $NO_3^-$ were determined using ion chromatography. Details on the methods of the $PM_{2.5}$ filter
analysis during the PANACEA winter and summer campaign can be found in (Kaskaoutis et al., 2022). In situ-
measurements of temperature and relative humidity were also available for the period of the campaigns (Fig S1
and S2).
For the campaign period there were not available gas phase concentrations ($NH_3$, $HNO_3$) measurements at any
of the 6 sampling sites except PTR ($NH_3$). Thus, for $NH_3$, satellite data were used for the remaining sites (except
IOA during the winter campaign period). For $HNO_3$, past in situ measurements were used at FKL and for the other
sites (THI, PTR, IOA, LAP, XAN) past in situ measurements conducted at THI were used.
In more detail, the Patras site as mentioned above, was the only site with available gas-phase $NH_3$
measurements for the studied periods during the 2019-2020 winter campaign, with mean ammonia concentration
of $2.54 \pm 0.90$ μg/m³. At IOA during the wintertime period of the study, neither in situ observations nor satellite
data were available and thus $NH_3$ observations at PTR were used for IOA. Therefore, the mean ammonia
concentration value from PTR dataset was also used for the IOA wintertime simulations.
For the other sites (FKL and THI during winter and summer, and LAP and XAN during winter) $NH_3$ data
(level 2 data, version 1.6.3) were used from the Cross-track Infrared Sounder (CrIS) instrument which is deployed
on board the Suomi National Polar-orbiting Partnership (SNPP) platform (into an orbit with an altitude of 824 km
above the Earth surface) (Shephard and Cady-Pereira, 2015). Ammonia values over these sites were obtained
from the CrIS Fast Physical Retrieval (CFPR) product, which provides $NH_3$ concentration data for a total of 15
vertical levels, with the value closest to the ground being taken as the near-surface concentration. From the



available CrIS near-surface NH$_3$ concentrations within a 50-km diameter area around each site, the value within
the pixel at the closest distance to the site is retained (Shephard et al., 2020; Shephard and Cady-Pereira, 2015).
The wintertime near-surface NH$_3$ concentrations as derived from CrIS for FKL, THI, XAN and LAP are shown
in Fig. 2. For summer at FKL and THI, mean values of summertime near-surface NH$_3$ (from 13/08/2019 to
19/08/2019 for FKL and from 13/08/2019 to 31/08/2019 for THI) 1.24 ± 0.61 µg/m$^3$ and 1.32 ± 1.53 µg/m$^3$
respectively, were used due to the scarcity of summertime data from CrIS in 2019.
For IOA during summer, a mean NH$_3$ concentration of 1.04 µg/m$^3$ was used as derived from the Atmospheric
Infrared Sounder (AIRS) aboard NASA's Aqua satellite (Level 3 data) operating from September 2002 to August
2016 (Warner et al., 2016) (the mean summertime 2002-2016 concentration was used).
For HNO$_3$ and HCl gas-phase concentration measurements, samples were collected only at Finokalia (2015 -
2016) using glass fiber filters coated with Na$_2$CO$_3$ and were analyzed by ion chromatography. The median of
these gas-phase measurements of HNO$_3$ and HCl were used here (0.63 µg/m$^3$ and 0.98 µg/m$^3$ for winter and 0.95
µg/m$^3$ and 1.34 µg/m$^3$ for summer, respectively). For all other sites, gas-phase measurements of HNO$_3$ conducted
at Thissio from 12/2014 to 3/2016 (Liakakou et al., 2022) were used here as mean values (0.53 ± 0.12 µg/m$^3$ for
winter and 0.91 ± 0.29 µg/m$^3$ for summer).
Since the NH$_3$ in situ observations at PTR did not coincide with any CrIS or AIRS data, a comparison between
in situ and satellite data was not possible. The sensitivity of our pH estimates to these assumed NH$_3$ and HNO$_3$
concentrations will be examined in a subsequent section (see section 3.4).

2.3 pH Estimation
The pH of the fine aerosol at the 6 studied sites, was calculated using ISORROPIA-lite thermodynamic model
(Kakavas et al., 2022), which is an extension of the ISORROPIA-II model (Fountoukis and Nenes, 2007).
ISORROPIA-lite treats the same system of aerosols as ISORROPIA-II (Ca$^{2+}$, K$^+$, Mg$^{2+}$, SO$_4^{2-}$, Na$^+$, NH$_4^+$, NO$_3^-$,
Cl$^-$, H$_2$O and their equilibrium with the gas phase HNO$_3$, NH$_3$, HCl and H$_2$O) with the addition of the organic
aerosol and considering only the deliquescent aerosol at all RH values. The addition of organic matter in the
thermodynamic equilibrium aerosol system, results in more aerosol water which favors the partitioning of
semivolatile inorganic species into the aerosol phase in order to satisfy the equilibrium (Kakavas et al., 2022).
The particle water (W$_{org}$) associated with the organic aerosol is implemented in the model using the hygroscopicity
parameter (κ$_{org}$). The total aerosol water content, i.e. the water associated with the inorganic and organic parts of
the aerosol, is the sum of the inorganic and organic water and it is used in the thermodynamic calculations. The
aerosol pH is then calculated by:
$$\text{pH} = -\log_{10} \frac{1000\, \gamma_{H^+}\, [\text{H}_{air}^+]}{[W_{inorg}] + [W_{org}]}$$

(1)

where $\gamma_{H^+}$ is the activity coefficient of the hydronium ion (H$^+$) here assumed unity, H$^+_{air}$ is the equilibrium particle
hydronium ion concentration per volume air (µg/m$^3$), W$_{inorg}$ and W$_{org}$ is the water associated with the inorganic
and organic part of the aerosol, respectively (both in µg/m$^3$). Thus, the aerosol pH was calculated including the
contribution of the organic aerosol hence the aerosol water associated with both inorganic and organic species.



To calculate the aerosol water content associated with the organic species, the total organic aerosol
concentration (OA) derived from the OC measurements at each site is used together with an OA/OC ratio of 1.8
for all sites. This average ratio is consistent with the results of measurements conducted at the same sites (Florou
et al., 2017; Hildebrandt et al., 2011; Kaskaoutis et al., 2020, 2022; Kostenidou et al., 2015; Pikridas et al., 2013;
Stavroulas et al., 2019; Tsiflikiotou et al., 2019). The hygroscopicity parameter was set to $\kappa_{org}$= 0.16 for Finokalia
as suggested in studies for this site (Bougiatioti et al., 2009; Kalkavouras et al., 2019), and $\kappa_{org}$=0.12 for the other
sites (Psichoudaki et al., 2018).
**3 Results and Discussion**
3.1 Aerosol Composition
The chemical composition and mass concentration of the major species in $PM_{2.5}$ measured during the PANACEA
campaigns are summarized in Table 1. The dominant $PM_{2.5}$ component was OA at all sites except FKL, where
sulfate dominated both in winter and summer. This is typical for remote background sites (Lemou et al., 2020;
Sciare et al., 2008). Higher sulfate levels were present in FKL during summer due to high temperatures and
photochemistry, followed by organics (typically consisting 20-30% of the $PM_{2.5}$ mass, Pikridas et al., 2010; Putaud
et al., 2004). The highest levels of OA were observed at Ioannina (IOA) during winter. OA represented more than
60% of the $PM_{2.5}$ and was mainly due to residential wood burning emissions while the meteorology and
topography of the area facilitated the accumulation of all pollutants (Kaskaoutis et al., 2022). Much lower levels
of OA and $PM_{2.5}$ mainly of regional origin, were measured at IOA in the summer (Kaskaoutis et al., 2022). At the
three sites where measurements were available in the summer (IOA, FKL, THI), sulfate was a major component
of the $PM_{2.5}$ despite their different characteristics. Inorganic $PM_{2.5}$ across all sites in winter was dominated by
sulfate except IOA where, interestingly, $NO_3^-$ was the dominant inorganic $PM_{2.5}$ component due to possible high
$NO_x$ emissions, or $NO_x$ accumulation in the boundary layer due to low inversion heights and decreased horizontal
circulation, the latter due to surrounding mountains. The $K^+$ levels in IOA were also elevated indicating the
influence of biomass burning in the area (Kaskaoutis et al., 2022).

3.2 Aerosol pH across Greece
3.2.1 Winter
The distributions of estimated pH of $PM_{2.5}$ as derived from the ISORROPIA-lite model for the 6 studied sites
during winter (January 2020) is shown in Fig. 3a.
In January the aerosols at THI were slightly acidic with a wide range of pH values (2.17 to 4.17), reflecting
the variation of the $PM_{2.5}$ composition and meteorology. The mean pH was $3.30 \pm 0.48$ (median 3.34). Days with
highly acidic aerosol, i.e. pH below 2.5, were associated with elevated sulfate levels and northerly winds. At FKL
the mean aerosol pH was $3.25 \pm 0.37$ and ranged from 2.73 to 3.89 (median 3.30). The elevated pH coincided
with high concentrations of nonvolatile cations (NVC) and $NH_3$ and the lower pH values (less than 2.8) occurred
during periods with high levels of sulfates. In the northern part of Greece, at XAN the pH range covered 2.24
units with a mean value of $2.81 \pm 0.53$ (median 2.69). The shift between acidic and moderately acidic conditions





was associated with changes mainly in $NH_3$ and sulfate levels. Days when pH reached almost 4, were characterized
by high potassium in combination with high OA and EC levels and low sulfates suggesting significant influence
of biomass burning. At LAP the pH of $PM_{2.5}$ ranged from 2.72 to 3.41, with a mean value of $3.01 \pm 0.31$. Please
note that only 4 days of $PM_{2.5}$ measurements were available in LAP during January 2020. The fine aerosol in IOA
had the lowest overall acidity with a mean value of $4.08 \pm 0.42$ units and a range between 3.55 and 5.14 units.
These levels of acidity are a consequence of the intense biomass burning during this winter campaign at IOA
(Kaskaoutis et al., 2022). Elevated $K^+$ levels were observed, due almost entirely to wood burning for heating.
Comparing IOA with the other sites in January, there is a statistically significant difference in aerosol acidity
levels due to higher $Ca^{2+}$, $K^+$ and $NH_3$ combined with lower temperature, despite the high sulfates levels in the
area (Table 1). At PTR aerosol pH ranged from 2.82 to 4.44 units with a mean value of $3.70 \pm 0.45$. NVCs ($Ca^{2+}$,
$K^+$, $Mg^{2+}$ and $Na^+$), $NH_3$ and sulfates drove the variability of the aerosol pH most of the days. Aerosols at LAP
and XAN exhibited the highest acidity (lowest pH levels) among all studied sites, while IOA and PTR exhibited
the highest aerosol pH levels (Fig. 3a). Remarkably, aerosol acidity at the urban background site (THI) was similar
to that at the coastal background site (FKL). On the other hand, the suburban site (PTR) had higher aerosol pH
mainly due to the likely higher $NH_3$ levels at PTR than THI and FKL (Table 1), considering though the different
ways $NH_3$ was measured for these sites. Considering the two remote rural and coastal areas (XAN and FKL), their
fine aerosol pH differed by about half a pH unit due to increased $Na^+$, $Mg^{2+}$ and relative humidity levels at FKL
together with slightly higher sulfate concentrations at XAN.

286        The accuracy of the aerosol pH predictions using ISORROPIA-lite was evaluated by comparing the observed

with the predicted partitioning coefficient of $NH_4^+$ ($\varepsilon(NH_4^+)$) and this comparison was possible only at PTR during
winter when both $NH_3$ and $NH_4^+$ were measured. A useful way to assess the reliability and potential uncertainties
of such models' predictions of aerosol pH is the comparison between measured and simulated gas-particle
partitioning fractions of semivolatile species (Guo et al., 2016). Figure 4 shows the comparison of the observed
and predicted $\varepsilon(NH_4^+)$ calculated for the case of PTR in winter. We focus on PTR since it was the only site where
$NH_3$ in-situ measurements were available making it a more suitable case for such comparison. The measurements
and predictions of total ammonia are well correlated ($R^2=0.78$) although the model underestimates by 30% the
$\varepsilon(NH_4^+)$. A sensitivity test and a re-evaluation of the model discussed in the supplementary section A3 (Fig S3)
indicates that this small underestimation can be attributed to the uncertainty in the nitric acid levels. Sensitivity
tests regarding the gas phase $NH_3$ and $HNO_3$ used to estimate aerosol pH are also provided (section 3.4). Overall,
$PM_{2.5}$ during winter varied acidic to moderately acidic with an average pH across all studied urban and urban
background sites of $3.57 \pm 0.44$. The overall $PM_{2.5}$ pH range across all 6 sites was from 1.72 to 5.14.

3.2.2 Summer
The seasonal variation of the acidity of $PM_{2.5}$ was investigated at three of the sites, THI, FKL and IOA in which
measurements were available for the summer 2019 PANACEA campaign. The mean $PM_{2.5}$ composition is
summarized in Table 1. During the whole period, $PM_{2.5}$ at the three sites was acidic with a mean pH of $1.35 \pm$
0.18 at THI, $1.75 \pm 0.62$ at IOA and $2.08 \pm 0.45$ pH at FKL (Figure 3c). At THI the $PM_{2.5}$ pH was consistently
low throughout the summer period with a minimum value of 1.13, and a maximum of 2. At IOA and FKL the
$PM_{2.5}$ pH was slightly higher than at THI. At FKL relative humidity levels increased the aerosol water content





and together with the relatively high $K^+$ and $Na^+$ levels led to higher aerosol pH. Higher $Ca^{2+}$ and $K^+$ levels at IOA
and slightly higher sulfate levels at THI seemed to be the factors controlling the difference in the aerosol pH
between these two sites. Overall, comparing the summertime aerosol pH levels at the three sites (Fig. 3c and S5),
a uniformity can be observed with high aerosol acidity being the case on most of the days, dropping even below
0 at IOA as a result of increased temperature and sulfate levels and reduced aerosol water. FKL and IOA had
similar aerosol pH levels since most of the major aerosol components had similar concentrations at the two sites.
The higher $Ca^{2+}$ at IOA was balanced by the higher $Na^+$ in FKL.

3.2.3 Impact of organics
In order to investigate the potential effect of OA on the aerosol pH levels, ISORROPIA-lite was also run
considering only the inorganic components of the aerosol (i.e. setting the OA concentration to zero). The mean
difference between the aerosol pH calculated considering all aerosol components, including organics and the one
accounting only for the inorganic content of $PM_{2.5}$ is shown in Figs. 3b and d for winter and summer, respectively.
The water associated with the OA should reduce the $H^+$ concentration and therefore increase the aerosol pH
(Kakavas et al., 2022). However, this expected increase in aerosol pH due to OA was not always found at all sites.
In winter at IOA, THI and PTR, the addition of aerosol water associated with the OA did not always increase
the overall aerosol pH. In fact, a mean decrease of about 0.3 pH units was found at IOA, which had the highest
OA concentrations among the 3 sites. Figure 5a depicts the aerosol water associated only with the inorganics
along with the total one (i.e. including the OA water) and relative humidity and in Fig 5b the timeseries of the
aerosol pH at IOA is shown in which again the one associated only with the inorganics and the one including the
OA water (total aerosol pH) is depicted. The expected behavior of the aerosol pH as a result of the addition of the
organics (an increase in aerosol pH is expected) is clear only at the end of January (from 26-01 to 29-01); the total
aerosol pH was higher than the one when OA is absent (by 0.19 pH units on average). This was due to the high
aerosol water content, which resulted from high relative humidity (above 80%). The addition of the OA increased
aerosol water even more (Fig 5a). The decrease of aerosol pH when OA was present, was observed in all the other
days in January where the addition of OA did not raise the aerosol water to levels that would result in increased
pH. The concentration of $H^+$ increased as a result of the addition of OA components, resulting in a more acidic
aerosol (lower aerosol pH). The highest decrease in pH (mean decrease of 0.58 pH units) was observed in days
where already relatively high aerosol pH occurred (inorganic aerosol pH above 4.5 units), while for the other
cases the decrease was smaller (0.18 units mean decrease). Furthermore, the addition of OA affected the
partitioning of the semivolatile species with a decrease in the gas phase (up to 0.10 μg/m³ for $NH_3$, 0.18 for $HNO_3$
and 0.12 for HCL), and a similar increase in the corresponding aerosol phase. At the other two sites the aerosol
pH decreased by about 0.08 pH units at THI and 0.01 at PTR. At FKL, XAN and LAP a small increase in aerosol
pH was found; 0.001, 0.05, 0.05 pH units increase respectively. In the summer, the pH increased at all sites when
considering OA water, with the highest predicted increase of about 0.15 at IOA. At the other sites the aerosol pH
increased by about 0.09 units at FKL and 0.11 at THI. Therefore, the effect of the organics on the pH at all sites
was on average less than 0.3 units.

3.2.4 Main factors controlling seasonality of pH





A clear seasonal difference in acidity was observed at all sites. In summer PM$_{2.5}$ at FKL was 1 pH unit more acidic
than in winter. To determine the main drivers of aerosol pH seasonal variability, a series of sensitivity simulations
were performed using the concentrations of aerosol components and meteorological conditions observed in winter
and replacing each time one of them by its corresponding mean value in summer. Then, the difference in the
estimated aerosol pH was calculated. The factors that were tested were chosen based on their difference between
summer and winter and were: temperature, sulfates, TNH$_3$ (sum of gas-phase NH$_3$ and particulate NH$_4^+$) and K$^+$.
The results for FKL are shown in Fig. 6f and Table S4 as ΔpH, that is the pH of the base case simulation minus
that of the perturbation scenario. Sulfate was the main contributor to the change in aerosol acidity between summer
and winter at FKL, where the 2 μg m$^{-3}$ concentration difference between the two seasons resulted in 1.66 units of
pH difference. Total ammonia availability was the second most important factor with a mean ΔpH of 0.39
followed by temperature (mean ΔpH =0.30) and K$^+$ (mean ΔpH = 0.15).
At IOA, aerosol average pH levels differed by 2.26 pH units between summer and winter with higher aerosol
pH in winter (Fig. 6a). Simulations were performed using ISORROPIA-lite for nine factors: temperature, relative
humidity, TNO$_3$ (sum of gas-phase HNO$_3$ and particulate NO$_3^-$), TNH$_3$, SO$_4^{2-}$, Na$^+$, K$^+$, Ca$^{2+}$ and OA. The results
are shown in Fig. 6b and Table S5. Temperature (ΔpH=0.76) and the availability of total nitric acid (ΔpH=0.98)
and ammonia ((ΔpH=0.59) had the greatest influence on the seasonal aerosol pH difference. The effect of K$^+$
(ΔpH=0.53), sulfates (ΔpH=0.45) and Ca$^{2+}$ (ΔpH=0.43), followed by organics (0.22), RH (0.21) and Na$^+$ (0.06)
were also notable.
THI also exhibited a significant seasonal difference in the aerosol acidity with PM$_{2.5}$ being approximately 2.6
times more acidic in summer than in winter (Fig. 6c). Eight sensitivity tests were conducted for the effect of:
temperature, relative humidity, SO$_4^{2-}$, K$^+$, Ca$^{2+}$, Na$^+$, TNH$_3$ and TNO$_3$ on pH (Table S6, Fig. 6d). Sulfate was
found to be the main contributor to the more acidic conditions in summer at THI with a mean absolute difference
in pH (winter - summer), ΔpH, of 2.8 units, followed by temperature and TNH$_3$ with absolute ΔpH of 0.42 and
0.48 units, respectively. The other factors contributed less to the seasonal difference in aerosol pH, with an
absolute ΔpH varying between 0.12 and 0.33 units.

3.3 Aerosol water
ISORROPIA-lite can also be used to calculate the contribution of each inorganic salt and organic fraction to the
total aerosol water content. Fig. 7 depicts the average contributions of the various inorganic salts and of OA to the
total aerosol water for all sites in winter (Fig. 7a) and summer (Fig. 7b).
In winter (Fig. 7a), OA contributed 60% to the total aerosol water at IOA and 55% at PTR. These were the
two sites with the highest levels of OA mainly due to residential wood burning. At LAP and THI, OA was also
the dominant aerosol component contributing to total aerosol water 37% and 33% respectively, while at XAN
sulfate salts were associated with 42% and OA with 32% of the water. Focusing on the inorganic components
alone, aerosol water was found to be controlled by nitrate and sulfate. Nitrate had the highest contributions to
inorganic aerosol water at IOA (24%), THI (31%), LAP (30%) and PTR (21%). NH$_4$NO$_3$ was the dominant salt
at IOA, LAP and PTR; while at THI NaNO$_3$ contributed 25% to the total water associated with inorganics along
with NH$_4$NO$_3$ (22%). The situation at FKL was different with chloride and sulfate being the major contributors to
aerosol water (40% and 36% respectively), while OA had only a minor contribution (5%) due to its very low





concentration at this site. The dominant chloride salt at FKL was $NH_4Cl$ contributing 31% to the total inorganic
water while $Na_2SO_4$ was the major sulfate salt, contributing 17%.
In summer in the three sites with available observations, sulfate dominated  aerosol water (Fig. 7b) with
contributions ranging from 70% at IOA to 80% at FKL and THI. $(NH_4)_2SO_4$ was the dominant salt at all three
sites, contributing more than half of the total water content at THI (58%) and 38% and 40% at IOA and FKL,
respectively. The second most dominant aerosol component was OA with 19% contribution to the total aerosol
water at THI, 27% at IOA and 14% at FKL.

3.4 Sensitivity of pH to $NH_3$ and $HNO_3$
Studies with ISORROPIA-II (Guo et al., 2015, Weber et al., 2016) have shown that neglecting gas-phase $NH_3$ in
the thermodynamic equilibrium calculation of pH results in an underestimation of at most one pH unit (Bougiatioti
et al., 2016). Due to the lack of in situ measurements of the gas phase $NH_3$ (except PTR) and $HNO_3$ during the
campaign periods, a sensitivity test was conducted varying the assumed concentrations of the gases that we used
for the aerosol pH estimation. In detail, simulations with half and double the concentrations of $NH_3$ and $HNO_3$
were conducted for all sites during both periods (winter and summer). The resulted difference in aerosol pH ($\Delta$pH)
due to these altered gas phase concentrations are depicted in Fig. 8 and the mean values of $\Delta$pH for each simulation
are summarized in Table S7 along with the concentrations of the gases for the base case scenarios (i.e. the
concentrations used for the main simulations). For the case of $NH_3$, an average (across all sites and both periods)
of 0.25 increase in aerosol pH was observed when double the amount of $NH_3$ introduced to each system and the
increase in pH ranged between 0.13 and 0.34 pH units. On the other hand, when half the amount of $NH_3$ was used
a smaller change was observed across the sites; the mean decrease in aerosol pH was -0.19 pH units while it
ranged from -0.07 to -0.31 units. For the sensitivity simulations varying the amount of $HNO_3$ the overall change
in aerosol pH was much less distinct. The mean decrease in the pH of the aerosol was between 0.01 and 0.15 pH
units when the concentration of $HNO_3$ was doubled (the overall mean decrease was 0.06). Using half the
concentration of $HNO_3$ resulted in a mean increase ranging from 0.01 to 0.15 pH units.

3.5.1 Sensitivity of PM levels to ammonia and nitric acid availability
We used the framework developed by Nenes et al. (2020) that relates the levels of aerosol pH with the formation
of aerosol nitrate (and ammonium), to investigate the sensitivity of the different aerosol systems studied here to
the gas-phase concentrations of $NH_3$ and $HNO_3$. The main parameters used in this framework that control the
secondary inorganic particulate matter sensitivity are the aerosol pH, liquid water content and temperature. The
conceptual idea is that there is a "sensitivity window" of pH levels in which the partitioning of nitrate shifts from
nitrate being predominately gaseous to mostly in the aerosol phase. When acidity is below this pH sensitivity
window, particulate nitrate is almost non-existent and consequently aerosol levels are insensitive to $HNO_3$
availability. In this case, aerosol reduction policies that only target nitric acid reduction cannot be effective (Nenes
et al., 2020), as there is no nitrate in the aerosol phase. Based on these criteria, this framework defines
characteristic levels of aerosol acidity at which the aerosol becomes insensitive to $NH_3$ (or $HNO_3$) levels and vice
versa. Therefore, four possible regimes of PM sensitivity can be derived; i) neither $NH_3$ nor $HNO_3$ are important
for PM formation, or PM formation is dominated by ii) $HNO_3$, iii) both $NH_3$ and $HNO_3$, and iv) $NH_3$ alone.





Figure 9 shows the PM$_{2.5}$ sensitivity maps derived using this framework and the daily data at all stations in
January 2020 (Fig. 9a) and at FKL, THI and IOA in summer (Fig. 9b) (34 days in total; in July and August 2019).
Considering the average temperature across the 3 sites (Table S8) the pH sensitivity window for each of the two
gases (NH$_3$ and HNO$_3$) was calculated as a function of aerosol water (LWC). The data that were used as well as
the aerosol pH and water from the combination of the datasets from the 3 sites are given in Table S8.
During winter, an overall sensitivity of PM$_{2.5}$ to HNO$_3$ is found as most of the points of the sites reside above
the blue line thus an increase in HNO$_3$ concentration will lead to its partitioning to the aerosol phase as nitrate.
On the other hand, most points are also above the red line indicating that PM$_{2.5}$ is most insensitive to gas phase
NH$_3$. Depending on the conditions and aerosol acidity levels, each site may have some days that deviate from the
HNO$_3$-only sensitivity region. At FKL out of the 17 days where HNO$_3$ sensitivity dominated, on 4 days PM$_{2.5}$ was
sensitive to both HNO$_3$ and NH$_3$ levels due to the slightly more acidic particles present. At THI PM$_{2.5}$ was sensitive
to HNO$_3$ on all 27 days that were examined, except for two days in which NH$_3$ sensitivity regime also occurred
and characterized by lower aerosol pH values (2.05 and 2.13 pH units). At IOA as a consequence of the lower
aerosol acidity, PM$_{2.5}$ was (as expected) almost exclusively in the HNO$_3$ sensitive regime where NH$_3$ mass
variations would not affect PM concentration. At IOA, NH$_3$ availability was found to play a role in aerosol
formation in two days when the aerosol water was very high in combination with the aerosol pH being slightly
lower. Regions with more acidic particles i.e. XAN and LAP were also in some cases in the HNO$_3$ and NH$_3$
sensitive region.
During summer PM$_{2.5}$ shifted out of the HNO$_3$ only sensitive region at all three sites and NH$_3$ started to play
a more important role for the aerosol levels due to the more acidic conditions compared to winter. In many days
at IOA and some at FKL the sensitivity to HNO$_3$ and the insensitive regime also seemed to exist when extremely
low aerosol water content was present. At THI PM$_{2.5}$ sensitivity showed the least "dispersed" picture compared
to the other sites, and PM$_{2.5}$ was exclusively sensitive to NH$_3$ availability due to the consistently high aerosol
acidity conditions. Consequently, for the studied period the inorganic PM$_{2.5}$ levels at THI would be reduced by
controlling NO$_x$ emissions in winter, and NH$_3$ controls in summer. However, sulfate is the major inorganic PM$_{2.5}$
component during the summer so the ammonia reductions would have a relatively small effect on the total fine
PM.

3.5.2 Importance of semivolatiles for deposition
Aerosol acidity and water effects on the partitioning of HNO$_3$ and NH$_3$ are linked with the total deposition of these
species, in both their gas and particulate forms. As HNO$_3$ and NH$_3$ are deposited about 10 times faster than their
particulate forms (NO$_3^-$, NH$_4^+$), the partitioning between gas and particulate phase affects how fast HNO$_3$/ NO$_3^-$
or NH$_3$/ NH$_4^+$ are deposited (Nenes et al., 2021) and therefore the distance to which they are transported within
the atmosphere before deposition (Baker et al., 2021). This gas-to-particle partitioning depends on the pH of the
aerosol water and on the LWC of the aerosol (Guo et al., 2015). Therefore, "fast" and "slow" deposition regimes
for HNO$_3$ and NH$_3$ can be defined as a function of aerosol pH and LWC. Figure 10 shows the reactive nitrogen
(sum of TNO$_3$ and TNH$_3$) deposition regimes at the sites, where data were available both in winter (Fig. 10a) and
summer (Fig. 10b), calculated using the average temperatures of the datasets for each season. Looking at the
characterization of the deposition domains for all the studied sites during winter, a main difference can be observed





in terms of the nitrogen deposition velocity (Fig. 10a). At all sites, $NH_3$ seems to always experience fast deposition.
On the other hand, nitrate's deposition rate varies together with each ability for transport to other areas. IOA and
PTR cases are almost exclusively in the $HNO_3$ slow – $NH_3$ fast regime due to the higher aerosol pH levels that
enhance $HNO_3$ partitioning to the particulate phase. The other sites are characterized by high deposition rates for
both $NH_3$ and $HNO_3$. In summer, the deposition is similarly characterized as fast for both $NH_3$ and $HNO_3$ at FKL,
THI and IOA due to the higher acidity levels. The higher temperature in summer than in winter also favors the
fast removal of both $HNO_3$ and $NH_3$, with the exception of one day at IOA characterized by extremely low aerosol
pH (below zero), where $NH_3$ was present in the form of particulate $NH_4^+$ and thus had a slow deposition velocity.

**Conclusions**
This study was based on $PM_{2.5}$ chemical composition observations in 6 regions of Greece (Finokalia, FKL;
Thissio, THI; Patras, PTR; Ioannina, IOA; Xanthi, XAN; and Thessaloniki, LAP) during summer 2019 and winter
2019-2020 as part of the national research infrastructure PANACEA. The aerosol composition measurements
together with the gas phase $NH_3$, $HNO_3$ data were used in the thermodynamic model ISORROPIA-lite to calculate
the fine aerosol pH at all the sites and to determine its seasonal variation.
The pH levels of $PM_{2.5}$ across Greece during winter ranged from 1.72 to 5.14. The highest pH values were
estimated at IOA ($4.08 \pm 0.42$) and PTR ($3.70 \pm 0.45$) followed by THI ($3.30 \pm 0.48$) and FKL ($3.25 \pm 0.37$),
while aerosols at XAN and LAP were the most acidic ($2.81 \pm 0.53$ and $3.01 \pm 0.31$, respectively). The lowest
acidity at IOA was associated with high $K^+$ levels from biomass burning emissions in combination with high $Ca^{2+}$
and $NH_3$ and low temperatures. Similar factors ($NH_3$ and cation levels) affected PTR aerosol pH. The aerosol
acidity levels at the urban background site (THI) were similar to those at the coastal background site (FKL). High
nonvolatile cation concentrations at FKL, together with elevated humidity levels resulted in a higher mean aerosol
pH compared to the rural background site (XAN), which had slightly higher levels of sulfates than FKL.
In summer, $PM_{2.5}$ was generally more acidic than in winter at the three sites studied (THI, IOA and FKL),
with mean pH values of $1.35 \pm 0.18$ (THI), $1.75 \pm 0.62$ (IOA) and $2.08 \pm 0.45$ units (FKL). pH in summer was
lower than winter by 1 (FKL) to about 2 (THI and IOA) pH units. Sulfates drive the seasonal variation in aerosol
pH at THI and FKL. At IOA, on the other hand, temperature together with $NH_3$ and $HNO_3$ availability were the
main drivers of the seasonal difference in aerosol acidity, with $K^+$, $Ca^{2+}$, and sulfates also contributing.
Organics contributed significantly to the total fine aerosol mass at all sites but FKL in winter. OA was found
to be the main contributor to the total aerosol water at IOA and PTR (60% and 55% contribution, respectively)
due to the OA levels in these sites, and the dominant contributor at THI and LAP (33% and 37% contribution
respectively). During winter nitrate salts contributed more to the total aerosol water at IOA (24%), THI (31%),
LAP (30%) and PTR (21%), with $NH_4NO_3$ being the dominant salt present at IOA, LAP and PTR and $NaNO_3$ at
THI. The aerosol water content at XAN was dominated by sulfate (42%), with OA also contributing to the total
aerosol water (32%). Chloride and sulfate contributed more to the aerosol water at FKL (40% and 36%
respectively). During summer, sulfate salts contributed more to LWC at all sites (80% contribution at FKL and
THI and 70% at IOA), with $(NH_4)_2SO_4$ being the dominant species of the inorganic LWC at all three sites. At
sites with higher aerosol pH (IOA, THI and PTR in winter), the water associated with the organics did not increase
aerosol pH in most cases.



PM$_{2.5}$ mass was sensitive to the availability of total HNO$_3$ at all sites during winter. At LAP and XAN, PM
mass was also found to be sensitive to NH$_3$. In summer, PM at all three sites examined showed a strong sensitivity
to NH$_3$ due to the low summertime aerosol pH. In some cases, the PM2.5 concentrations at FKL and IOA appeared
insensitive to both precursor species due to the low water content of the aerosols. PM sensitivity at THI in summer
showed a clear dependence on NH$_3$, reflecting the higher summertime aerosol acidity. Our results show that HNO$_3$
levels (could contribute to) regulate PM$_{2.5}$ mass concentration which however was mainly composed by OA and
sulfate, hence policies targeted to reduce PM$_{2.5}$ levels in Greece would be more effective by reducing NOx
emissions (i.e. transportation sector) in addition to OA and sulfate.
Finally, our analysis has shown that in Greece NH$_3$ deposition is fast, whereas deposition of HNO$_3$/NO$_3^-$ may
occur locally near the sources or remotely by long-range transport, depending on the environmental conditions.
How future changes in the meteorological conditions and in air pollutant emissions will affect the aerosol pH and
the factors controlling it, as well as atmospheric residence time and deposition of reactive nitrogen in the region,
requires further investigation.

Acknowledgements. We acknowledge support of this work by the project "PANhellenic infrastructure for
Atmospheric Composition and climatE change" (MIS 5021516) which is implemented under the Action
"Reinforcement       of       the       Research       and       Innovation       Infrastructure"
(http://www.antagonistikotita.gr/epanek_en/proskliseis.asp?id=28&cs=) , funded by the Operational Programme
"Competitiveness, Entrepreneurship and Innovation" (NSRF 2014-2020) and co-financed by Greece and the
European Union (European Regional Development Fund), by the European Research Council (ERC-2016-COG),
Project Pyrogenic TRansformations Affecting Climate and Health (PyroTRACH-726165) by the Horizon-2020
Project FORCeS of the European Union under grant agreement No 821205 and by the Horizon-2020 project
Research Infrastructures Services Reinforcing Air Quality Monitoring Capacities in European Urban & Industrial
AreaS (RI-URBANS) grant agreement No. 101036245.

Colorblind friendly palettes were used to create the figures according to and described in (Crameri et al., 2020).
More information about the abovementioned palettes can be found in https://doi.org/10.5281/zenodo.8409685
(Crameri, 2018).

Code  availability.  ISORROPIA-lite  is  openly  available  at  https://www.epfl.ch/labs/lapi/models-and-
software/isorropia/

Data availability. Observational data are available upon request by the corresponding site principal investigator.

Competing interests. Some authors are members of the editorial board of journal Atmospheric Chemistry and
Physics



Author contributions. AMN collected the data, performed the simulations and wrote the manuscript, MK
conceived the study and supervised the work. MK and SP edited the manuscript, AN provided advice on the use
of ISORROPIA model, and the frameworks by Nenes et al (2020 and 2021). MT, EL, KP, KT, IT, CK, FB, AK,
KM, GK, NK, NH provided data; all authors provided comments on the manuscript.

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





FIGURES AND TABLES

| | |
|---|---|
| 728 | |
| 729 | |
| 730 | |
| 731 | |
| 732 | |
| 733 | |
| 734 | |
| 735 | |
| 736 | |
| 737 | |
| 738 | |
| 739 | |
| 740 | |
| 741 | |
| 742 | |
| 743 | |

**Figure 1: Map of the region of Greece and the sampling sites. From south to north, Finokalia (FKL), Thissio (THI), Patras (PTR), Ioannina (IOA), Thessaloniki (LAP) and Xanthi (XAN). Original satellite image of Greece obtained from (https://upload.wikimedia.org/wikipedia/commons/9/92/Satellite_image_of_Greece.jpg )**

**Table 1: Descriptive statistics (mean ± stdev) for the chemical composition of PM$_{2.5}$ during the winter and summer PANACEA campaigns. *OA* is the *organic aerosol* derived from OC measurements and a ratio of O$A$/OC of 1.8. Meteorological conditions are also provided. (*) The aerosol pH and LWC calculated with ISORROPIA-lite are also provided (n= number of days).**

| Observations | FKL | | THI | | PTR | IOA | | LAP | XAN |
|---|---|---|---|---|---|---|---|---|---|
| µg/m³ | winter | summer | winter | summer | winter | winter | summer | winter | winter |
| OA | 0.69 ± 0.49 | 2.91 ± 1.10 | 4.77 ± 4.53 | 7.51 ± 5.27 | 17.35 ± 7.61 | 50.36 ± 34.46 | 5.06 ± 1.77 | 12.45 ± 7.97 | 6.78 ± 3.33 |
| Na$^+$ | 0.39 ± 0.25 | 0.46 ± 0.33 | 0.24 ± 0.11 | 0.17 ± 0.13 | 0.32 ± 0.30 | 0.19 ± 0.11 | 0.12 ± 0.06 | 0.46 ± 0.49 | 0.14 ± 0.38 |
| NH$_4$$^+$ | 0.20 ± 0.25 | 0.99 ± 0.37 | 0.40 ± 0.25 | 1.46 ± 0.50 | 0.67 ± 0.43 | 1.12 ± 0.81 | 0.79 ± 0.33 | 1.04 ± 0.76 | 0.75 ± 0.58 |
| K$^+$ | 0.18 ± 0.24 | 0.25 ± 0.13 | 0.29 ± 0.24 | 0.12 ± 0.05 | 0.75 ± 0.43 | 1.45 ± 1.00 | 0.27 ± 0.18 | 0.37 ± 0.17 | 0.30 ± 0.16 |
| Ca$^{2+}$ | 0.27 ± 0.40 | 0.18 ± 0.14 | 0.23 ± 0.24 | 0.23 ± 0.20 | 0.34 ± 0.44 | 0.96 ± 0.78 | 0.55 ± 0.25 | 0.33 ± 0.11 | 0.17 ± 0.06 |
| Mg$^{2+}$ | 0.10 ± 0.12 | 0.06 ± 0.03 | 0.02 ± 0.02 | 0.06 ± 0.01 | 0.03 ± 0.04 | 0.03 ± 0.02 | 0.04 ± 0.01 | 0.04 ± 0.01 | 0.04 ± 0.02 |
| SO$_4$$^{2-}$ | 1.30 ± 0.59 | 3.71 ± 1.18 | 0.77 ± 0.43 | 4.42 ± 1.59 | 1.78 ± 0.90 | 2.77 ± 1.42 | 3.27 ± 1.02 | 2.05 ± 1.00 | 1.90 ± 1.36 |
| NO$_3$$^-$ | 0.30 ± 0.30 | 0.14 ± 0.11 | 0.66 ± 0.50 | 0.16 ± 0.09 | 1.39 ± 0.78 | 3.66 ± 2.73 | 0.17 ± 0.07 | 2.01 ± 1.71 | 0.70 ± 0.58 |
| Cl$^-$ | 0.37 ± 0.47 | 0.31 ± 0.30 | 0.40 ± 0.19 | 0.40 ± 0.12 | 0.25 ± 0.27 | 0.61 ± 0.38 | 0.26 ± 0.07 | 0.33 ± 0.14 | 0.40 ± 0.18 |
| T (°C) | 11.83 ± 2.74 | 24.84 ± 1.66 | 10.64 ± 3.22 | 28.97 ± 1.90 | 11.14 ± 2.36 | 7.43 ± 2.52 | 26.86 ± 2.60 | 9.94 ± 2.65 | 8.62 ± 2.90 |
| RH (%) | 72.03 ± 9.68 | 65.82 ± 9.66 | 65.85 ± 11.65 | 46.04 ± 6.69 | 64.66 ± 11.97 | 68. 06 ± 20.25 | 51.59 ± 9.73 | 57.62 ± 13.17 | 62.43 ± 16.88 |
| *Month (number of samples)* | January (n=16) | July & August (n=34) | January (n=27) | July & August (n=34) | January (n=16) | January (n=27) | July & August (n=34) | January (n=4) | January (n=26) |
| pH* | 3.25 ± 0.37 | 2.08 ± 0.37 | 3.30 ± 0.48 | 1.38 ± 0.18 | 3.70 ± 0.45 | 4.08 ± 0.42 | 1.82 ± 0.65 | 3.01 ± 0.31 | 2.81 ± 0.53 |
| LWC (µg/m³)* | 6.85 ± 3.65 | 5.85 ± 3.05 | 3.06 ± 2.96 | 3.34 ± 1.62 | 7.00 ± 5.36 | 56.61 ± 127.59 | 1.97 ± 0.79 | 29.68 ± 32.67 | 4.57 ± 4.29 |

| | |
|---|---|
| 744 | |
| 745 | |
| 746 | |
| 747 | |
| 748 | |
| 749 | |
| 750 | |
| 751 | |
| 752 | |
| 753 | |



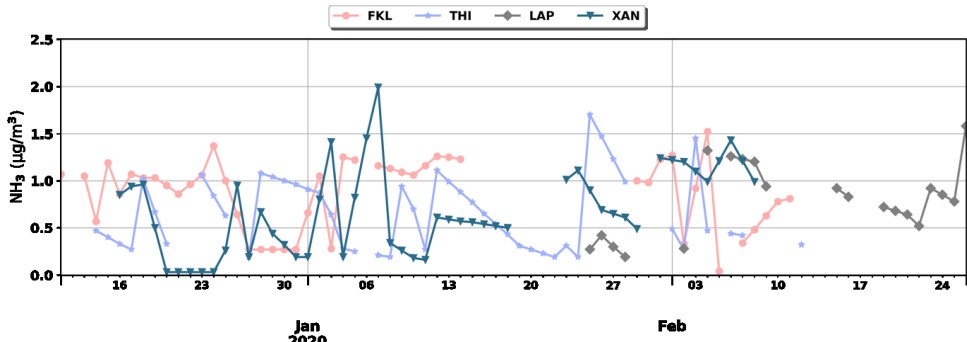


**Figure 2: Daily surface NH₃ concentrations during the winter of 2019-2020 as derived from the Cross-track Infrared**
**Sounder (CrIS) instrument for FKL, THI, LAP and XAN.**



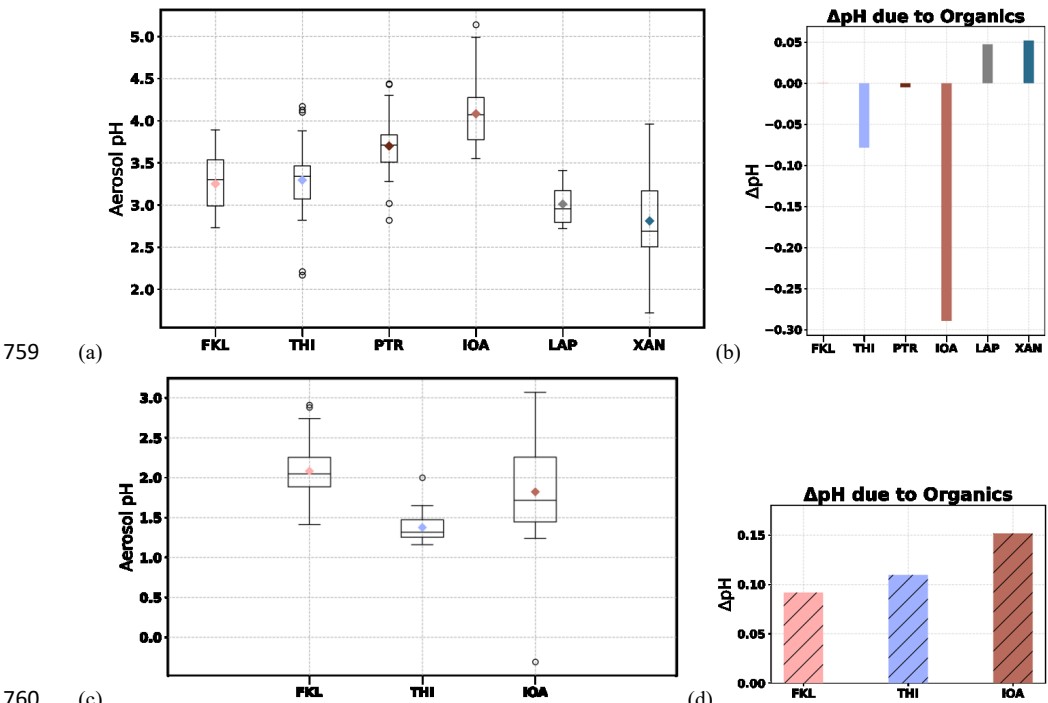


**Figure 3: a) and c) total aerosol pH (meaning the aerosol pH associated with inorganics and organics) derived using**
**ISORROPIA-lite in January 2020 and summer 2019 respectively, b) and d) ΔpH = total aerosol pH – inorganic aerosol**
**pH both derived from ISORROPIA-lite in January 2020 and summer 2019 respectively. The inorganic aerosol pH was**
**derived by setting the organic concentration and hygroscopicity as zero.**



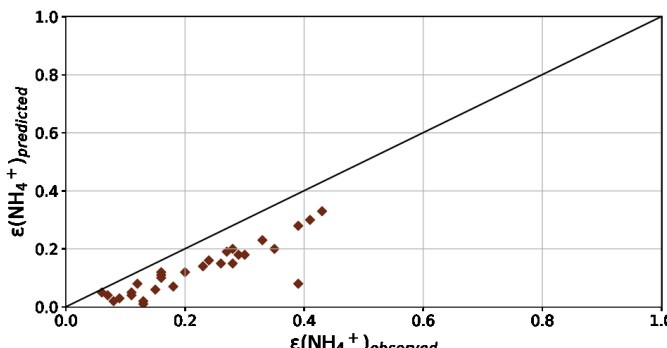


**Figure 4: Evaluation of the ISORROPIA-lite results for the case of PTR in winter. Comparison of the partitioning coefficient of NH$_4^+$ calculated from measurements with that from the predicted concentrations derived from the model (R$^2$ = 0.78 and y = 0.70x - 0.03). The partitioning coefficient is defined as ε(NH$_4^+$) = NH$_4^+$/(NH$_4^+$ + NH$_3$).** *The black line shows the 1:1 ratio.*

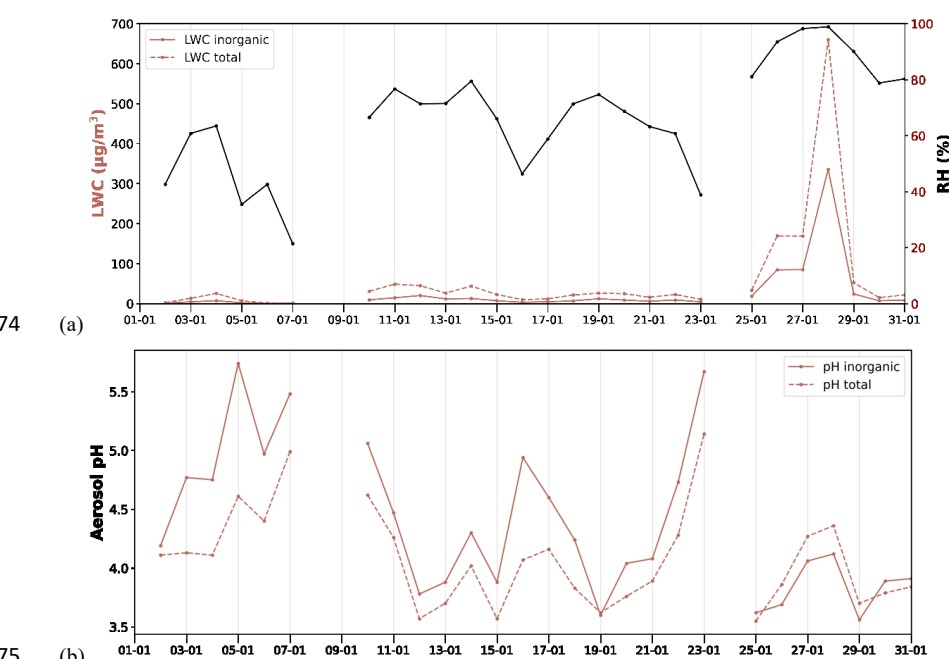

**Figure 5: (a) Aerosol liquid water content at IOA in January 2020 as derived from ISORROPIA-lite associated with inorganics (inorganic, *red line*), associated with both inorganics and organics (total, *red- dashed line*) and relative humidity levels (secondary y axis, *black line*). (b) Inorganic *(red line)* and total (*red dashed line,* associated with inorganics and organics) aerosol pH at IOA in January 2020.**





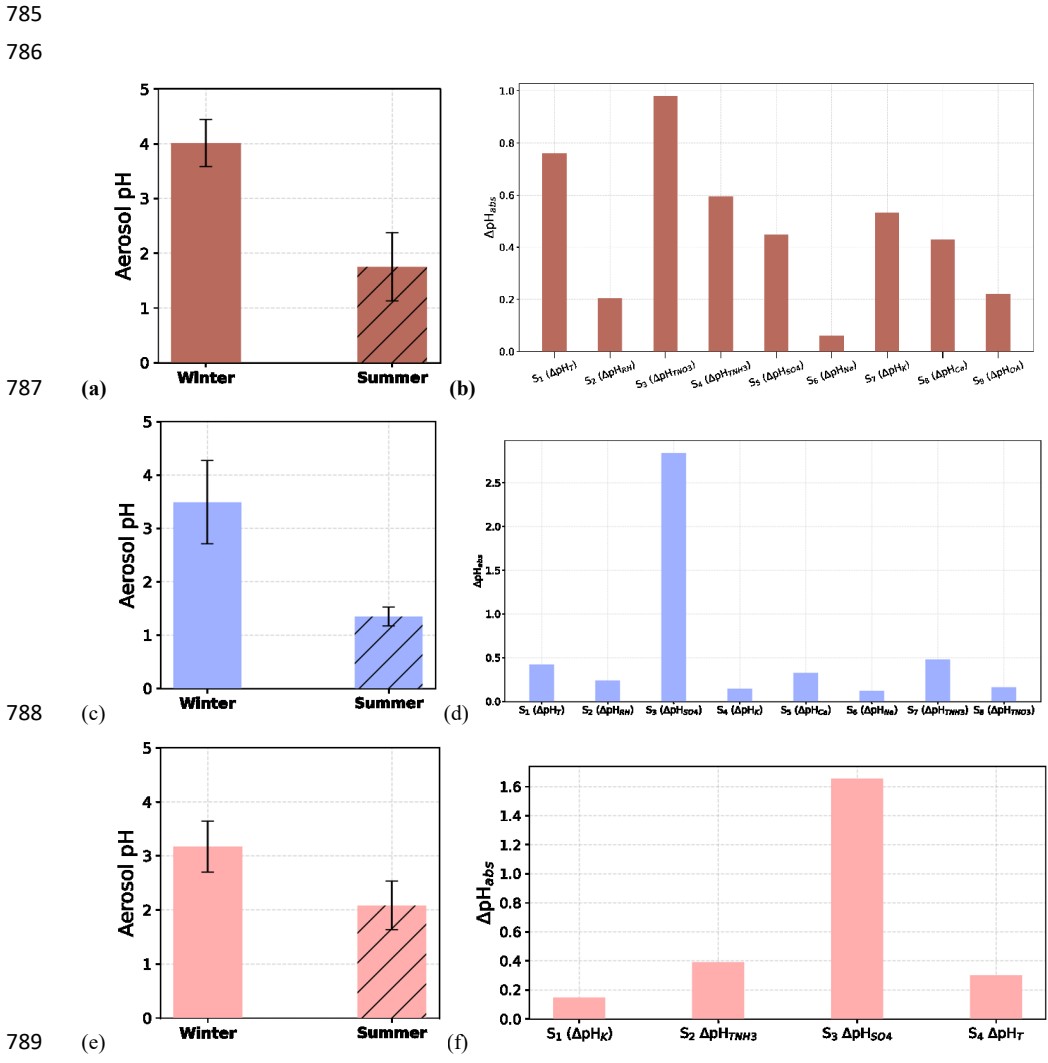

(a)          (b)
(c)          (d)
(e)          (f)
**Figure 6: Mean fine aerosol pH in winter and in summer at (a) IOA (c) THI, (e) FKL and the respective sensitivity**
**results for winter conducted for each site (Tables S4-S6). $\Delta pH_{abs} = |pH_{winter} - pH_{winter(summerX)}|$, where summerX is (b)**
**for IOA: the mean summer temperature (first simulation, $S_1$), relative humidity ($S_2$), $TNO_3$ ($S_3$), $TNH_3$ ($S_4$), $SO_4^{2-}$ ($S_5$),**
**$Na^+$ ($S_6$), $K^+$ ($S_7$), $Ca^{2+}$ ($S_8$) and $OA$ concentration ($S_9$) (d) for THI: the mean summer temperature ($S_1$), relative humidity**
**($S_2$), $SO_4^{2-}$ ($S_3$), $K^+$ ($S_4$), $Ca^{2+}$ ($S_5$), $Na^+$ ($S_6$), $TNH_3$ ($S_7$) and $TNO_3$ ($S_8$) concentration and (e) for FKL: the mean summer**
**$K^+$ ($S_1$), $TNH_3$ concentration ($S_2$), $SO_4^{2-}$ concentration ($S_3$) and temperature ($S_4$).**



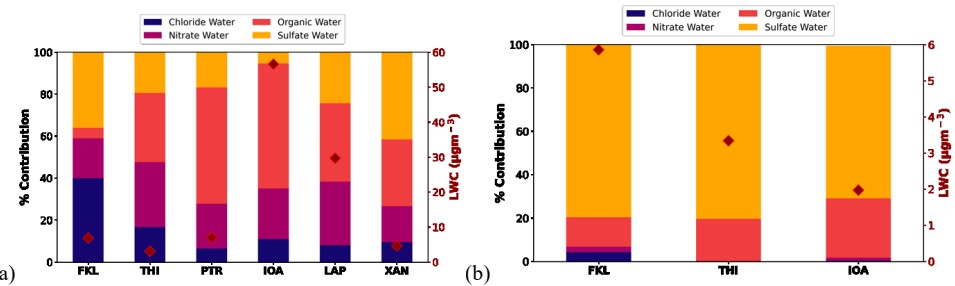

(a)                                                                (b)

**Figure 7:** Averaged aerosol liquid water content (LWC) for all sites in winter, January 2020 (a) and summer, July and August 2019 (b) expressed as the contribution of each chemical aerosol salt group considering both the inorganics and organics at the observational sites shown in Fig. 1 (left axis). Aerosol water mean concentrations are shown with rhombus in each plot (right axis).

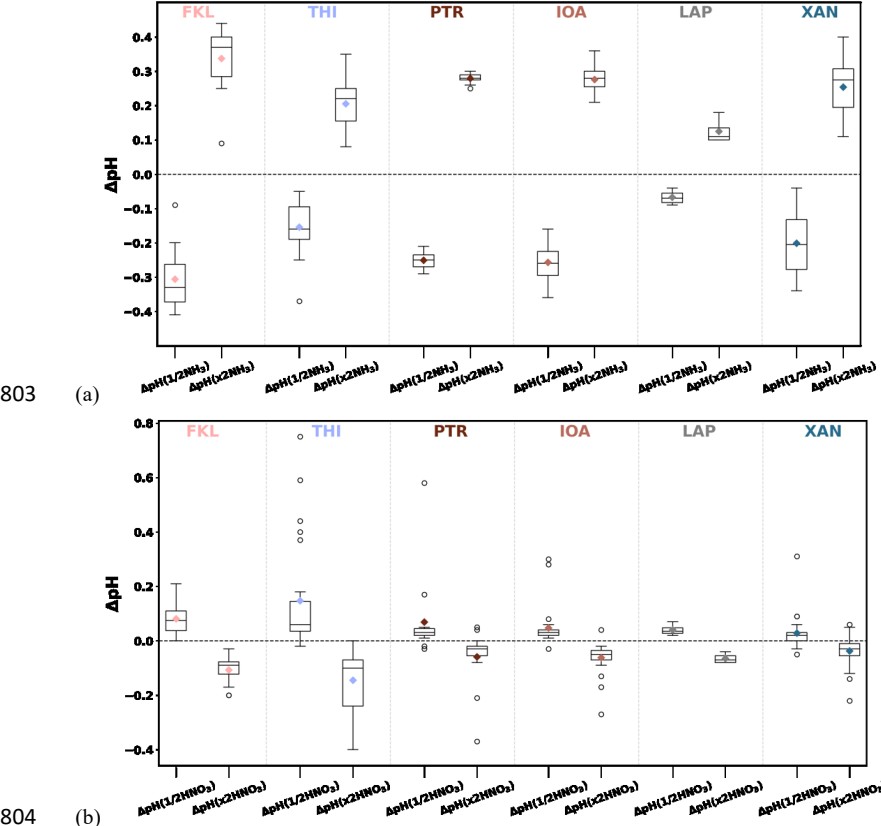

(a)

(b)



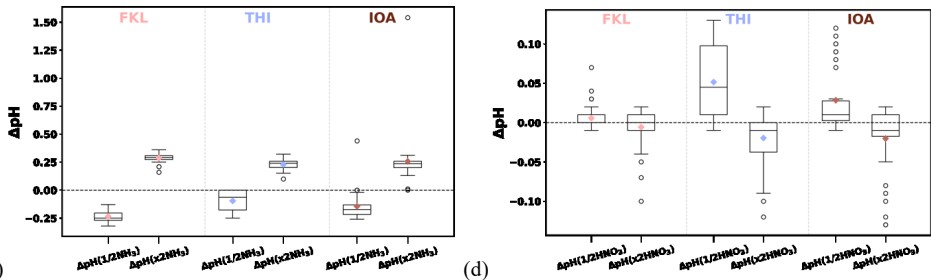

(c)                     (d)

**Figure 8: Sensitivity of pH estimation to gas phase NH₃ and HNO₃ concentrations. (a) and (b) are the wintertime (January) simulations for NH₃ and HNO₃ tests respectively and (c) and (d) for the summertime ones. Each graph shows the difference in pH when using half or double the concentration of the respective gas compared to the original pH at each site. (ΔpH = pH$_{1/2gas}$ − pH$_{original}$ and ΔpH = pH$_{x2gas}$ − pH$_{original}$ respectively).**

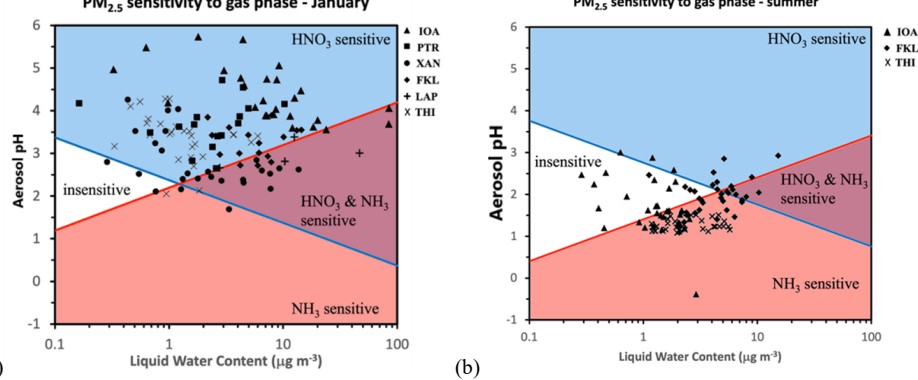

(a)                     (b)

**Figure 9: Chemical domains of sensitivity of PM$_{2.5}$ *mass* to NH₃ and NO$_x$ emissions for the studied period (a) in winter (January) and (b) in summer (July and August). The average temperature used here is the mean measured one for the studied period at all sites in each season. Daily averaged values in each season of aerosol pH and liquid water content were used. *The red line shows the characteristic aerosol pH as a function of liquid water content below which the PM$_{2.5}$ mass is sensitive to NH₃ levels and the blue line the characteristic aerosol pH as a function of liquid water content above which the PM$_{2.5}$ mass is sensitive to HNO₃ levels.***





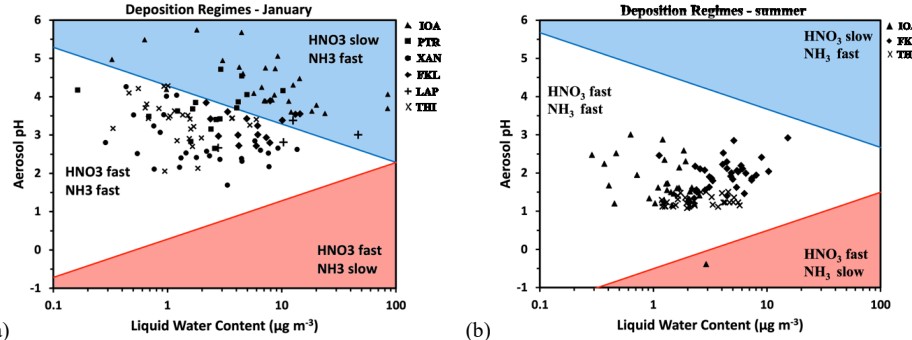

(a)                                                                                                    (b)

**Figure 10: Domains of reactive nitrogen deposition for the studied period (a) in winter (January) and (b) in summer (July and August). The average temperature used here is the mean measured one at all sites, for each season. Daily averaged values in each season for aerosol pH and liquid water content were used.** *The red line shows the characteristic aerosol pH, as a function of liquid water content, below which NH₃ deposition is slow and the blue line the characteristic aerosol pH, as a function of liquid water content, above which HNO₃ deposition is slow.*