# Peer review of "Spatial and temporal distribution of fine aerosol acidity in the"

_EGUsphere, 2025_

## Referee Comment (RC1)

The manuscript titled "Spatial and temporal distribution of fine aerosol acidity in the Eastern Mediterranean" by Neroladaki et al. characterizes aerosol acidity in Greece by comparing and contrasting measurements among six geographic sites and between summer and winter time periods. Using an aerosol thermodynamics model, the study finds that pH is lower during the summer than winter due mostly to sulfate variability, that organics do not always increase pH when aerosol pH is higher during the winter, that aerosol pH is more sensitive to $NH_3$ during the summer since pH is lower, and that $NH_3$ deposition is fast while $HNO_3$ deposition depends on environmental conditions. The study provides interesting insights and has potential to increase understanding on aerosol acidity, however, there are some areas that are unclear or lack necessary supporting information that should be addressed prior to publication. Following these revisions, I recommend publication.

**Specific Comments**
Line 112: Remove ", trend"

Line 185 (and throughout the manuscript): The text jumps back and forth between the full spelling of chemical species and their abbreviations. In one sentence starting on Line 184, "$NH_3$" is first used then "ammonia" is used several words later. For clarity and consistency of the manuscript, it is recommended to stick to only the abbreviations or only the full spellings.

Line 199: add "are" before "1.24..."

Section 2.2: For the gas-phase measurements, the manuscript utilizes a combination of simultaneous in-situ measurements at its site, simultaneous in-situ measurements at a neighboring site, past in-situ measurements at its site, past in-situ measurements at a neighboring site, and satellite measurements at its site that are averaged over 1 year at some sites and over 14 years at another. While the text at present explains the various measurements, it is very confusing and difficult to keep track of this information (which is relevant to understanding the results and impact of the study). It would be beneficial to create a table that summarizes the type of gas-phase measurements (satellite vs. in-situ, simultaneous vs. past, same site or neighboring site) for each of the 6 sites. This would allow for a simplified comparison of the measurements used and can serve as quick reference for the reader while going through the results section. Such a table may be included in the supplement and referenced from the main text.

Section 2.2: It is not clearly stated in the methods that summer measurements are not available at all 6 sites. Table 1 is the first (indirect) mention that PTR, XAN, and LAP only have winter measurements. The time frame of measurements at each site should be explicitly stated in the methods. Additionally, it is suggested that measurements are during the summer of 2019 and winter of 2019-2020. Later in the results section, it is mentioned that only January 2020 measurements are used. January 2020 would be winter 2020 not winter 2019-2020. Please clarify in the methods section the specific measurement times, and use only one same term (either winter 2020 or January 2020) throughout the manuscript for consistency.

Line 205: Why is the median used for HNO3 and HCl at FKL while the mean was used for HNO3 at all other sites (and the mean was also used for satellite measurements)? Please justify the one use of the median while all other cases utilize the mean.

Section 3.2.4: Are the differences in pH between summer and winter statistically significant (e.g. by a two-sample t-test)?

Section 3.3: A total of 9 parameters were tested at IOA and THI but only 4 parameters were tested at FKL. This leaves RH, TNO3, Na, Ca, and OA missing at FKL. Why were these not run for FKL? Based on the averages listed in Table 1 these measurements are available and so there is not an apparent reason for why they were not considered at FKL. Please either add the 5 missing parameters for FKL or justify why they were not considered.

Section 3.4: What are the implications of the results of this sensitivity analysis specifically to the results of this manuscript? It seems that this sensitivity analysis is particularly relevant to the results at the IOA site since satellite measurements were used during the summer but in-situ measurements from PTR were used during the winter. The results of the change in aerosol pH between summer and winter (Section 3.2.4, Figure 6) then could possibly be muddled by differences in gas-phase measurements demonstrated in Section 3.4. Please discuss.

Figure 5a: The plot lines and axis labels match in color but the axis ticks are mis-matched in color.

Figure 6, 8: The axis labels are too small to be legible. Please make them larger or consider using different abbreviations.

---

## Author Comment (AC1)

We would like to thank the reviewer for the constructive comments and feedback on our manuscript. We considered all comments and have revised our manuscript accordingly. We provide below a point by point response to the comments. Reviewer comments are shown in *italic* and our responses follow.

**Reviewer 1**

**Comment 1:**

Line 112: Remove ",trend"

Response:

Removed.

**Comment 2:**

Line 185 (and throughout the manuscript): The text jumps back and forth between the full spelling of chemical species and their abbreviations. In one sentence starting on Line 184, "NH3" is first used then "ammonia" is used several words later. For clarity and consistency of the manuscript, it is recommended to stick to only the abbreviations or only the full spellings.

**Response:**

In the revised manuscript we first provide the full name and abbreviation of the chemical species and then only the abbreviation is used.

**Comment 3:**

Line 199: add "are" before "1.24..."

**Response:**

Added.

**Comment 4:**

Section 2.2: For the gas-phase measurements, the manuscript utilizes a combination of simultaneous in-situ measurements at its site, simultaneous in-situ measurements at a neighboring site, past in-situ measurements at its site, past in-situ measurements at a neighboring site, and satellite measurements at its site that are averaged over 1 year at some sites and over 14 years at another. While the text at present explains the various measurements, it is very confusing and difficult to keep track of this information (which is relevant to understanding the results and impact of the study). It would be beneficial to create a table that summarizes the type of gas-phase measurements (satellite vs. in-situ, simultaneous vs. past, same site or neighboring site) for each of the 6 sites. This would allow for a simplified comparison of the measurements used and can serve as quick reference for the reader while going through the results section. Such a table may be included in the supplement and referenced from the main text.

**Response:**

We thank the reviewer for this comment, which helped increase the clarity of our study. As suggested by the reviewer, we have added a summary table that provides the type of gas-phase measurements for each of the 6 sites. It is now included in the supplement as Table S2.

**Comment 5:**

Section 2.2: It is not clearly stated in the methods that summer measurements are not available at all 6 sites. Table 1 is the first (indirect) mention that PTR, XAN, and LAP only have winter measurements. The time frame of measurements at each site should be explicitly stated in the methods. Additionally, it is suggested that measurements are during the summer of 2019 and winter of 2019-2020. Later in the results section, it is mentioned that only January 2020 measurements are used. January 2020 would be winter 2020 not winter 2019-2020. Please clarify in the methods section the specific measurement times, and use only one same term (either winter 2020 or January 2020) throughout the manuscript for consistency.

**Response:**

In the revised manuscript, we now clarify that FKL, THI and IOA are the only sites with available summertime measurements. The campaigns took place in summer 2019 and winter 2019-2020. However, the timing of the measurements differs from site to site. Therefore, in order to be able to compare the aerosol pH among the sites for winter we only used January 2020 data, except in Section 3.2.4. In that section, where we investigated the seasonality of aerosol pH in the 3 sites, we used all available days with measurements in each site in order to include as many data points as possible, since we did not make a straight comparison among the sites and rather we investigate the seasonality in each site separately. We now use "January 2020" when only data from this month are used. Similarly, to properly compare the three sites in the summer, we used the common days with available measurements in July and August. Clarification is made in the revised version of the manuscript. For winter:

"Due to the lack of data in some of the sites during the winter 2019-2020, the results of only January 2020 were used in order to be able to compare the aerosol pH results among the sites. The entirety of the aerosol pH results for all sites during winter 2019-2020 can be found in Fig. S4." For summer:

"Overall, comparing the summertime (July and August) aerosol pH levels at the three sites (Fig. 3c and S5), a uniformity can be observed with high aerosol acidity being the case on most of the days, dropping even below 0 at IOA as a result of increased temperature and sulfate levels and reduced aerosol water."

**Comment 6:**

Line 205: Why is the median used for HNO3 and HCl at FKL while the mean was used for HNO3 at all other sites (and the mean was also used for satellite measurements)? Please justify the one use of the median while all other cases utilize the mean.

**Response:**

Indeed we should have used the mean concentrations of the gases in order to be consistent. However, we recalculated the aerosol pH and water using the mean values and found that there is no significant difference between using median values. The aerosol pH and water were not statistical different from the ones calculated with the mean values with a confidence level of 95% (student two sample t-test and Mann–Whitney U test). Below we provide the results using both mean and median concentrations. The winter and summer results are in parenthesis; (w) and (s) respectively.

```
With median values: Aerosol pH \rightarrow 3.25 \pm 0.37 (w), 2.08 \pm 0.37 (s)
Aerosol water \rightarrow 6.85 \pm 3.65 (w), 5.89 \pm 3.05 (s)
```

```
With mean values: Aerosol pH \rightarrow 3.22 \pm 0.34 (w), 2.08 \pm 0.45 (s)
Aerosol water \rightarrow 7.10 \pm 3.81 (w), 5.41 \pm 3.24 (s)
```

**Comment 7:**

Section 3.2.4: Are the differences in pH between summer and winter statistically significant (e.g. by a two-sample t-test)?

**Response:**

The differences in pH between summer and winter are indeed statistically significant with confidence level 95% in all 3 sites (FKL, THI, IOA). We use both a student two sample t-test and Mann–Whitney U test.

**Comment 8:**

Section 3.3: A total of 9 parameters were tested at IOA and THI but only 4 parameters were tested at FKL. This leaves RH, TNO3, Na, Ca, and OA missing at FKL. Why were these not run for FKL? Based on the averages listed in Table 1 these measurements are available and so there is not an apparent reason for why they were not considered at FKL. Please either add the 5 missing parameters for FKL or justify why they were not considered.

**Response:**

We used only 4 parameters in FKL instead of 9 as in IOA due to their insignificant effect on seasonality. The same effect occurred in THI, though. In order to be consistent, we repeated the sensitivity tests in FKL and in THI considering all 9 parameters. The simulations are now consistently named among the sites ( $S_1$  is always the simulation when temperature is tested,  $S_2$  when relative humidity is tested, etc.). The corresponding panels in Fig. 6 were replotted. We have modified the manuscript in section 3.2.4 and Tables S5 and S7 in the supplement accordingly.

**Comment 9:**

Section 3.4: What are the implications of the results of this sensitivity analysis specifically to the results of this manuscript? It seems that this sensitivity analysis is particularly relevant to the results at the IOA site since satellite measurements were used during the summer but in-situ measurements from PTR were used during the winter. The results of the change in aerosol pH between summer and winter (Section 3.2.4, Figure 6) then could possibly be muddled by differences in gas-phase measurements demonstrated in Section 3.4. Please discuss.

**Response:**

Indeed, the differences in the input data used motivated us to perform this sensitivity analysis. The results of the sensitivity analysis are now further discussed for IOA following the reviewer comment.

"According to this sensitivity analysis, the uncertainty in the gas phase NH3 could explain about half of the seasonal difference in aerosol pH that was presented and discussed in section 3.2.4 about the factors affecting the seasonality of pH. The uncertainty in gas phase NH3 could explain a larger fraction of the seasonal difference in aerosol pH in IOA than in FKL and THI, which is expected since at these two sites aerosol pH seasonality was predominately driven by sulfates. Our results show a factor of 2 uncertainty in NH3 lead to an average pH difference of 0.25 units that has minor impact on our findings discussed below."

**Comment 10:**

Figure 5a: The plot lines and axis labels match in color but the axis ticks are mis-matched in color.

**Response:**

We thank the reviewer for pointing out this mismatch that is now corrected in the revised figure.

Comment 11:
Figure 6, 8: The axis labels are too small to be legible. Please make them larger or consider using different abbreviations.

**Response:**

Indeed, the figures were not easily legible. We revised those figures and increased the size of the axis

---

## Author Comment (AC2)

We would like to thank the reviewer for the constructive comments and feedback on our manuscript. We considered all comments and have revised our manuscript accordingly. We provide below a point by point response to the comments. Reviewer comments are shown in *italic* and our responses follow.

**Reviewer 2**

**Main comments**

**Comment 1:**

The abstract and conclusions (line 481) emphasize  $K^+$  levels drove the higher pH/less acidic nature of aerosols at IOA. How was that effect identified? Is that based on previous knowledge or a specific calculation/test? The  $K^+$  as a driver of seasonality seems to be a separate conclusion supported by Figure 6 (line 490).

**Response:**

Indeed K+ importance as a driver of aerosol pH in IOA is supported by the seasonality sensitivity (Fig 6) but it is not the only factor affecting the aerosol pH. We based our conclusion on the comparison of the aerosol pH and K+ concentrations among the studied sites. In addition, Kaskaoutis et al. (2022) presented and analyzed the observations from IOA and found that residential wood burning was the dominant source of the aerosol in winter and that the accumulation process of the pollutants was further promoted by the meteorological conditions in the area.

The relevant text in the revised manuscript is now:

"The effect of K+ ( $\Delta$ pH=0.53), sulfates ( $\Delta$ pH=0.45) and Ca2+ ( $\Delta$ pH=0.43), followed by organics (0.22), RH (0.21) and Na+ (0.06) were also notable. High concentrations of K+ affecting the aerosol pH are associated with biomass burning as discussed in Kaskaoutis et al., (2022)."

**Comment 2:**

How do organics affect inorganic species in ISORROPIA-lite? Section 3.2.3: What interaction cause the unexpected decrease in pH when organics were removed? Are any of the organics dissociating acids? Does the organic water combine with the inorganic water for calculation of NHx and TNO3 partitioning?

**Response:**

The detailed description of ISORROPIA-lite is provided in Kakavas et al. (2022).

Briefly, the organic aerosol contributes additional aerosol water to the system beyond what is derived from inorganic species alone. The effect of the organic aerosol in ISORROPIA-lite is to increase the aerosol liquid water content, which in turn is allowed to affect the partitioning of semivolatile inorganics. Dissociation of organic acids is not considered in the thermodynamic equations. The particle water associated with the organic aerosol (Worg) is parameterized through the hygroscopicity parameter

$$W_{\text{org}} = \frac{m_{\text{org}}\rho_{\text{w}}}{\rho_{\text{org}}} \frac{k_{\text{org}}}{\left(\left(\frac{1}{RH}\right) - 1\right)}$$

where  $m_{org}$  is the organic mass (in  $\mu g/m^3$ ),  $\rho_{org}$  is the organic density (1.35 g/cm3),  $\rho_w$  is the density of water,  $k_{org}$  (0.12) is the organic hygroscopicity parameter and RH the relative humidity.

In IOA, during the last period of the campaign, when RH was higher than 80%, the increase in aerosol liquid water content due to organics was very large leading to lower H+ concentration in the aerosol water and therefore a higher aerosol pH. During the other days of the campaign, relative humidity was lower and the increase in aerosol liquid water content due to organics was not strong enough to substantially affect the aerosol pH.

This part of the discussion now reads as follows:

"The decrease in aerosol pH when OA was present, was observed on all the other days in January when the lower relative humidity and the addition of OA did not raise the aerosol water to levels that would result in increased pH. As a result of the addition of OA in the model, the concentration of H+ increased, as well as the associated aerosol water. However, the increase in aerosol water does not counterbalance the increase in H+ concentration resulting in a more acidic aerosol (lower aerosol pH)."

**Comment 3:**

Line 448, line 508, and elsewhere regarding NOx emissions and nitrate: Womack et al. (2019) show that nitrate formation can be NOx or VOC limited. Consider that VOC emissions may govern total nitrate abundance. Do you know if nitrate in the airshed(s) is more sensitive to NOx vs VOC controls?

**Response:**

Existing studies conducted in Greece/include Greece suggest that the nitrate formation is strongly correlated with local NOx emissions (over the greater Athens area, Myriokefalitakis et al., 2024). Reducing NOx and other anthropogenic emissions in the area of Athens decreases PM2.5 mainly by reducing secondary inorganic aerosols (including particulate nitrate) (Im and Kanakidou, 2012). Moreover, Tsimpidi et al. (2025) attributed observed declines in particulate nitrate in Europe over the past two decades largely to NOx emission reductions. Finally, Megaritis et al. (2013) found that the particulate nitrate over Greece is most likely sensitive to NOx emission controls and VOC controls have a much smaller impact.

These sentences have been modified as follows:

"Consequently, for the studied period the inorganic PM2.5 levels at THI in winter would be reduced by limitation of HNO3 formation, which depends on VOC and NOx conditions, and NH3 controls in summer."

"Our results show that  $HNO_3$  levels (could contribute to) regulate  $PM_{2.5}$  mass concentration which however was mainly composed by OA and sulfate, hence policies targeted to reduce  $PM_{2.5}$  levels in Greece would be more effective by reducing  $HNO_3$  levels (i.e. transportation sector) in addition to OA and sulfate."

**Comment 4:**

If a figure/analysis can be added to address the uncertainty introduced by the lack of complete gasphase NH3 and HNO3 measurements, that would be useful. That could inform future work by letting the community know how precise they likely need to with estimated NH3 and HNO3 concentrations if they need to fill in that data due to lack of measurements. Did the authors consider an iterative technique in which an initial guess of gas-phase NH3 and HNO3 was used to construct total NHx and TNO3 input to predict a new NH3 and HNO3 gas estimate and run until predicted aerosol NH4 and NO3 converged to measurements? Zheng et al. (2020) showed that the lack of NH3 gas values can lead to an incorrect direction in the pH trend over time. It is unclear what size error in NH3 is acceptable.

**Response:**

Regarding the initial conditions of the semi volatile species used in the model, gas phase NH3 and HNO3 were used. This input data originates from in situ, and satellite measurements of the gas phase, as well as in situ measurements of the particle phase (NH4+, NO3-). (see the revised supplementary Table S2).

The use of gas phase NH3 and HNO3 concentrations in the thermodynamic model is now clearly stated in Section 2.3.

"Daily values were used as input to the model. These were gas phase (NH3, HNO3) and particulate phase (ions and OA) concentrations, OA hygroscopicity and density, and meteorological data (temperature and relative humidity)."

In section 3.4 we address the impact of gas phase NH3 and HNO3 uncertainty on aerosol pH calculation. We found a 0.25 pH units change when NH3 was doubled and -0.19 when NH3 was halved. This sensitivity is consistent with previous studies showing that a 10-fold change in NH3 results in one unit change of aerosol pH (Guo et al., 2017; Weber et al., 2016). For HNO3 a range between 0.01 to 0.15 pH units difference was found when half and double the amount of HNO3 was used.

In the revised manuscript we added the following discussion at the end of Section 3.4, also addressing a relevant comment of the other reviewer:

"According to this sensitivity analysis, the uncertainty in the gas phase NH3 could explain about half of the seasonal difference in aerosol pH that was presented and discussed in section 3.2.4 about the factors affecting the seasonality of pH. The uncertainty in gas phase NH3 could explain a larger fraction of the seasonal difference in aerosol pH in IOA than in FKL and THI, which is expected since at these two sites aerosol pH seasonality was predominately driven by sulfates. Our results show a factor of 2 uncertainty in NH3 lead to an average pH difference of 0.25 units that has minor impact on our findings discussed below."

Regarding the iterative technique, we did not apply such method in our study. However, we are aware that Ibikunle et al. (2024) examined if aerosol pH and gas-phase NH3 can be constrained from measurements of NH4+, total nitrate and nitrate partitioning, with promising results for aerosol pH. The same study examined other iterative approaches proposed in the literature and showed that they could lead to numerical instability.

**Minor comments**

**Comment 1:**

Line 86: Define "neutral levels".

**Response:**

We added in parenthesis the neutral pH level.

"Negative values appear when sulfates are the dominant constituent of particulate matter, while the aerosol pH rarely rises above neutral levels (pH = 7)."

**Comment 2:**

Section 2.3: What time averaging was used for the input conditions to ISORROPIA-lite?

**Response:**

We thank the reviewer for this comment. Daily values were used as input in the model.

For clarity, we added the following sentence to the revised manuscript in section 2.3 (pH estimation):

"Daily values were used as input to the model. These were gas phase (NH3, HNO3) and particulate phase (ions and OA) concentrations, OA hygroscopicity and density, and meteorological data (temperature and relative humidity)."

**Comment 3:**

Line 360-361: Order the variable impacts from largest to smallest impact.

**Response:**

We changed the order and now the variables are mentioned from largest to smallest impact as suggested.

**The text now reads:**

"The availability of total HNO3 ( $\Delta pH=0.98$ ), temperature ( $\Delta pH=0.76$ ) and the total NH3 ( $\Delta pH=0.59$ ) had the greatest influence on the seasonal aerosol pH difference."

**Comment 4:**

Line 431: Change "most insensitive" to "relatively insensitive" or "least sensitive".

**Response:**

We changed "most insensitive" to "relatively insensitive".

**Comment 5:**

Line 464: check wording.

**Response:**

The following revision to this sentence was made: "On the other hand, nitrate's deposition rate varies between fast (as gaseous HNO3) and slow (as particulate NO3-), leading to local removal or long-range transport, respectively."

**Comment 6:**

*In several figures (Fig 3, 6, 8), subplot labels on the figure would be useful.*

**Response:**

We added the labels on the subplots as suggested.

**Comment 7:**

Does the inclusion of organics and their associated water substantially affect Figures 9-10?

**Response:**

We thank the reviewer for this comment. We included two figures (Fig. S6 and S7) in the supplement with the frameworks of Fig. 9 and 10 using the aerosol liquid water content calculated including the organics. In the main text we included the following comments:

**At the end of 3.5.1:**

"The inclusion of the OA in the aerosol pH and water calculations resulted in a small difference in terms of these sensitivity maps. As the aerosol water increased slightly, in a few cases in XAN the PM sensitivity was shifted from insensitive to HNO3 sensitive, while in a few cases in IOA and FKL, PM also became sensitive to NH3 (Fig. S6)."

**At the end of 3.5.2:**

"Including the OA in the calculations did not change as much the deposition rates (Fig. S7). In a few cases during January in THI and PTR the deposition rate of HNO3 shifted from fast to slow due to the addition of OA water."

**References**

Guo, H., Weber, R. J., and Nenes, A.: High levels of ammonia do not raise fine particle pH sufficiently to yield nitrogen oxide-dominated sulfate production, Sci Rep, 7, 12109, https://doi.org/10.1038/s41598-017-11704-0, 2017.

Ibikunle, I., Beyersdorf, A., Campuzano-Jost, P., Corr, C., Crounse, J. D., Dibb, J., Diskin, G., Huey, G., Jimenez, J.-L., Kim, M. J., Nault, B. A., Scheuer, E., Teng, A., Wennberg, P. O., Anderson, B., Crawford, J., Weber, R., and Nenes, A.: Fine Particle pH and Sensitivity to NH3 and HNO3 over South Korea During KORUS-AQ, Chimia, 78, 762–770, https://doi.org/10.2533/chimia.2024.762, 2024.

Im, U. and Kanakidou, M.: Impacts of East Mediterranean megacity emissions on air quality, Atmos. Chem. Phys., 12, 6335–6355, https://doi.org/10.5194/acp-12-6335-2012, 2012.

Kakavas, S., Pandis, S. N., and Nenes, A.: ISORROPIA-Lite: A Comprehensive Atmospheric Aerosol Thermodynamics Module for Earth System Models, Tellus B: Chemical and Physical Meteorology, 74, 1–23, https://doi.org/10.16993/tellusb.33, 2022.

Kaskaoutis, D. G., Grivas, G., Oikonomou, K., Tavernaraki, P., Papoutsidaki, K., Tsagkaraki, M., Stavroulas, I., Zarmpas, P., Paraskevopoulou, D., Bougiatioti, A., Liakakou, E., Gavrouzou, M., Dumka, U. C., Hatzianastassiou, N., Sciare, J., Gerasopoulos, E., and Mihalopoulos, N.: Impacts of severe residential wood burning on atmospheric processing, water-soluble organic aerosol and light absorption, in an inland city of Southeastern Europe, Atmospheric Environment, 280, 119139, https://doi.org/10.1016/j.atmosenv.2022.119139, 2022.

Megaritis, A. G., Fountoukis, C., Charalampidis, P. E., Pilinis, C., and Pandis, S. N.: Response of fine particulate matter concentrations to changes of emissions and temperature in Europe, Atmos. Chem. Phys., 13, 3423–3443, https://doi.org/10.5194/acp-13-3423-2013, 2013.

Myriokefalitakis, S., Karl, M., Weiss, K. A., Karagiannis, D., Athanasopoulou, E., Kakouri, A., Bougiatioti, A., Liakakou, E., Stavroulas, I., Papangelis, G., Grivas, G., Paraskevopoulou, D., Speyer, O., Mihalopoulos, N., and Gerasopoulos, E.: Analysis of secondary inorganic aerosols over the greater Athens area using the EPISODE–CityChem source dispersion and photochemistry model, Atmos. Chem. Phys., 24, 7815–7835, https://doi.org/10.5194/acp-24-7815-2024, 2024.

Petters, M. D. and Kreidenweis, S. M.: A single parameter representation of hygroscopic growth and cloud condensation nucleus activity, Atmos. Chem. Phys., 11, 2007.

Tsimpidi, A. P., Scholz, S. M. C., Milousis, A., Mihalopoulos, N., and Karydis, V. A.: Aerosol composition trends during 2000–2020: in-depth insights from model predictions and multiple worldwide near-surface observation datasets, Atmos. Chem. Phys., 25, 10183–10213, https://doi.org/10.5194/acp-25-10183-2025, 2025.

Weber, R. J., Guo, H., Russell, A. G., and Nenes, A.: High aerosol acidity despite declining atmospheric sulfate concentrations over the past 15 years, Nature Geosci, 9, 282–285, https://doi.org/10.1038/ngeo2665, 2016.

---

## Author Response (AR2)

We thank the reviewer for allowing us to further improve the clarity of our manuscript. We considered the comment, have revised the section 3.2.3 accordingly and added a relevant sentence to the conclusion. Please find our reply here below. We also submitted a track changes manuscript and the supplement with the technical corrections asked by editorial office. We hope that this revised version is now suitable for publication in ACP.

**Reply to reviewer**

*Comment:*
*I have reviewed responses to first round reviews and found items to be adequately addressed. In the case of Section 3.2.3, additional insight into the increase in acidity due to uptake of water onto organics would be useful. For example, did the increased water cause more nitric acid uptake? Or, did NH3 in the gas-phase increase? Is there evidence of a discontinuity when organic water is added that could be leading to an artifact?*

Response:

Addition of organic aerosol water most of the times tends to reduce acidity and elevate pH, as shown by Kakavas et al. (2022) and other related studies. However, under conditions where organic aerosol water content can dominate the aerosol water however, it is possible that acidity can be increased. In the discussed case, the addition of organics and the subsequently increase of the aerosol water resulted in an increase in the partitioning of all three semi volatiles ($NH_3$, $HNO_3$ and $HCl$) from the gas phase to the aerosol phase. The gas phase $NH_3$, $HNO_3$ and $HCl$ concentrations decreased and the respective concentrations of the aerosol phase increased. Gas phase $NH_3$ did not increase. The addition of considerable amounts of water affected the activity coefficients of all ionic species (especially sulfate/bisulfate ions and $NH_4^+$) which shifted the ionic balance slightly. The resulting changes from the addition of OA are consistent with increases in aerosol water that depend on RH, changes in the partitioning of semivolatiles, the concentrations of which increased in the aerosol phase and decreased in the gas phase, as well as in changes in the activity coefficients of all ionic species. The increase in the concentrations of the acids exceeded that of ammonium resulting in a decrease of the pH. Note also that as shown in eq. 1 in the manuscript, we are using the $pH_F$ proxy for the pH (free $H^+$ approximation of pH that does not consider shifts in the activity coefficient in $H^+$; Pye et al., Atmos. Chem. Phys., 2020).

[revised manuscript text omitted]